# Scalability and some optimization of the Finite-volumE Sea-ice-Ocean Model, Version 2.0 (FESOM2)

Nikolay V. Koldunov[1,2], Vadym Aizinger[3,2], Natalja Rakowsky[2], Patrick Scholz[2], Dmitry Sidorenko[2], Sergey Danilov[2,4], and Thomas Jung[2]

[1]MARUM—Center for Marine Environmental Sciences, Leobener Str. 8, 28359 Bremen, Germany
[2]Alfred Wegener Institute, Helmholtz Centre for Polar and Marine Research, Am Handelshafen 12, 27570 Bremerhaven, Germany
[3]University of Bayreuth, Chair of Scientific Computing, 95447 Bayreuth, Germany
[4]Jacobs University, Campus Ring 1, 28759 Bremen, Germany

**Correspondence:** Nikolay V. Koldunov (nikolay.koldunov@awi.de)

**Abstract.** A study of the scalability of the Finite-volumE Sea-ice-Ocean circulation Model, Version 2.0 (FESOM2), the first mature global model of its kind formulated on unstructured meshes, is presented. This study includes an analysis of main computational kernels with a special focus on bottlenecks in parallel scalability. Several model enhancements improving this scalability for large numbers of processes are described and tested. Model grids at different resolutions are used on four HPC systems with differing computational and communication hardware to demonstrate the model's scalability and through-put. Furthermore, strategies for improvements in parallel performance are presented and assessed. We show that, in terms of throughput, FESOM2 is on a par with state-of-the-art structured ocean models and, in a realistic eddy-resolving configuration (1/10° resolution), can achieve about 16 years per day on 14 000 cores. This suggests that unstructured-mesh models are becoming very competitive tools in high-resolution climate modelling. We show that the main bottlenecks of FESOM2 parallel scalability are the two-dimensional components of the model, namely the computations of the external (barotropic) mode and the sea-ice model. It is argued that these bottlenecks are shared with other general ocean circulation models.

## 1  Introduction

Mesoscale eddies play a critical role in the general circulation of the ocean, strongly affect biogeochemical processes, and thus impact marine life. Since these eddies are not resolved on meshes coarser that the local internal Rossby radius, their effect must be parameterized in terms of eddy-driven bolus transport (Gent and McWilliams (1990)) or mixing. While such coarse-mesh models have been a staple of climate research, there are numerous indications summarized in Hewitt et al. (2017) that eddy-resolving meshes can enhance the realism of climate models by improving the simulated positions of major frontal systems and reducing surface and deep biases in climate simulations (Rackow et al., 2019). With increasing computational resources becoming available to the community, running eddy-permitting (nominal resolution around 1/4°) and eddy-resolving (resolution of 1/10° or better) simulations becomes more and more feasible in climate research. However, existing models are

still rather slow and require many months of wallclock time for century-scale climate simulations making these configurations prohibitively costly for many important applications.

The grid resolution requirements become even more severe in the modeling of many physical processes in the ocean. Since the first internal Rossby radius is decreasing in high latitudes down to several kilometers or even less on the ocean shelves, very fine meshes are needed to simulate the emerging eddy dynamics. At midlatitudes, simulating the submesoscale eddy dynamics related to mixed layer or ageostrophic instabilities, which are particularly pronounced in the vicinity of fronts, creates comparable challenges with respect to model resolution that, ideally, should be on the kilometer scale. Furthermore, submesoscale eddies are believed to affect mixed layer restratification and vertical heat transport (see, e.g., Sasaki et al. (2014), Su et al. (2018)) and may contribute to shaping the circulation in major current systems (Chassignet and Xu (2017)). Further challenges may emerge in designing setups capable of resolving simultaneously large-scale and coastal dynamics (Holt et al. (2017)). Studying their role in ocean puts very high demands on the efficiency of ocean circulation models.

Research questions such as how the ocean will impact climate in a warming world are the main drivers behind the ongoing work towards increasing numerical efficiency, and thus throughput, of ocean circulation models. However, even if ample computational resources were available, one important factor hampering the throughput of these models is their limited parallel scalability, that is, existing models struggle to make full use of the new generation of massively parallel HPC systems. Scalability bottlenecks often arise from the saturation of the parallel communication after mesh partitions become smaller than some number of surface mesh vertices per compute core depending on the model and on the hardware employed. The question of scalability is widely discussed in the ocean modeling community, yet the number of publications that document the present status is modest (see, e.g., Huang et al. (2016); Dennis et al. (2012); Prims et al. (2018); Ward and Zhang (2015); Kiss et al. (2019)). Furthermore, the information propagates largely via personal exchanges between researchers rather than through peer-reviewed publications. In fact, it can be argued that the opportunities and challenges that come with extreme-scale computing still miss the majority of researchers involved in code development. In order to establish the status quo and to raise awareness of the most pressing problems, a first necessary step is to determine precisely the limits of scalability of existing systems, analyse their common causes, and to understand to what extent the existing ocean circulation models are different.

The main components of ocean circulation models limiting their scalability have been identified in the literature as the solver for the external (barotropic) mode (e.g. Huang et al. (2016); Prims et al. (2018)) and the sea-ice model (e.g. Prims et al. (2018)). They represent two-dimensional (2D) stiff parts of the solution algorithm and require either linear solvers (usually iterative) or explicit pseudo-time-stepping with very small time steps (see the split-explicit method for the barotropic dynamic in Shchepetkin and McWilliams (2005) or elastic-viscous-plastic method for the sea-ice in Hunke and Dukowicz (1997)). Both approaches are not particularly computationally expensive; however, they introduce numerious exchanges of 2D halos per time step of the ocean model. Thus the extent to which the 2D parts of the code control the scalability and the measures potentially capable of at least partly alleviating these limitations are worth a detailed analysis. Typical numbers from the structured-mesh modeling community indicate that a 1/4° ocean mesh with about 1M wet surface vertices starts to saturate beyond 500 cores (see, e.g., Prims et al. (2018)) providing a throughput of 5–10 simulated years per day (SYPD). The throughput of the 1/10° Parallel Ocean Program (POP) model starts to saturate after 4000 cores at ca. 5 SYPD (∼5.8M wet surface vertices) increasing

to 10.5 SYPD at 16,875 cores with a new solver for the external mode (Huang et al. (2016)). Higher throughputs may be still possible but at the cost of a very inefficient use of computational resources.

The present manuscript considers the computational performance and parallel scalability of the unstructured-mesh Finite-volumE Sea-ice-Ocean circulation Model (FESOM2, Danilov et al. (2017)). FESOM2 is based on FESOM1.4 (Wang et al. (2014))—the first mature general-circulation model on unstructured mesh developed for climate research applications—but using a faster dynamical core. Recently developed large-scale ocean circulation models formulated on unstructured meshes such as FESOM (Wang et al., 2014; Danilov et al., 2017; Scholz et al., 2019), MPAS (Ringler et al., 2013; Petersen et al., 2019), or ICON (Korn (2017)) make it possible to utilize more flexible meshes with variable resolution. However, compared to structured-mesh models, the unstructured mesh models pose two specific issues that need to be addressed.

– First, mesh partitioning in unstructured mesh models is based on the analysis of the connectivity pattern between surface mesh cells or vertices; this method is agnostic of mesh geometry and allows one to work with only wet cells and vertices. FESOM2 uses the METIS software package (Karypis and Kumar (1998)) for this purpose. The number of neighboring parallel partitions is not defined *a priori* by quadrilateral geometry of the mesh as in many structured mesh models but may vary in some limits—due to the partitioning process—especially when mesh fragments become small. Hence, the performance of this technology on large global ocean meshes with complex boundaries and its implications for parallel communication are an interesting topic to explore.

– Second, one expects that an unstructured-mesh code is more expensive per degree of freedom than its structured-mesh counterparts due to indirect addressing of neighboring cells. However, for ocean meshes, this issue hardly causes any difficulties. In fact, due to the vertical alignment of 3D prisms within a column, the neighborhood information is propagated along the column and can be efficiently accessed. On the other hand though, high-order advection stencils on an unstructured mesh tend to be more complex and expensive to evaluate. Additionally, a triangular (or dual hexagonal) surface mesh contains more edges than a quadrilateral mesh with the same number of vertices making the computation of fluxes more expensive. This relative slowness can be compensated by better scalability leading to an equal or even higher throughput. Indeed, the number of halo exchanges and information to be exchanged should be about the same for meshes of the same size run on the same number of cores independently of their structure. Their relative expense measured as the communication-to-computation ratio will be lower, permitting scaling down to smaller partitions. It is therefore expected that unstructured-mesh codes can offer very similar throughput to that of structured-mesh models, and we see the substantiation of this statement as one of the main goals of this paper.

In the framework of the present study, we complement the analysis of scalability and throughput of FESOM2 as a whole by illustrating the performance of its main two- and three-dimensional computational kernels. The main scalability bottlenecks, such as the sea-ice and the solver for the external mode, are subject to an in-depth parallel performance analysis; the prospective and realized strategies to improve their performance are presented. The hierarchic mesh partitioning is an additional technology, which promises an improvement in the modularity and the flexibility of the main FESOM2 computational kernels and in their mapping to the current and prospective HPC systems. We introduce and discuss its possible uses in various contexts.

The paper is organized as follows. We start with the general description of the model and its solution algorithm, briefly introduce the meshes for test problems, and give some relevant information on the HPC systems employed in our study. In Sec. 3, the parallel performance results on our test systems are demonstrated separately for the whole model and for its main computational kernels—similarly to the approach used in Reuter et al. (2015). Sec. 4 includes some sample model analysis plots produced using Intel Trace Analyzer and deals with performance enhancements implemented in FESOM2. A number of additional tests illustrating various aspects of the model's performance constitute Sec. 5. A discussion of results and strategies for future improvements of the parallel scaling of FESOM2 is presented in Sec. 6, followed by a brief conclusions section.

## 2  Description of the model and test setups

### 2.1  Governing equations and solution procedure

FESOM2 (Danilov et al. (2017), Scholz et al. (2019)) is a general ocean circulation model solving the primitive equations in the Boussinesq, hydrostatic, and traditional approximations. It is formulated on unstructured triangular meshes with scalar degrees of freedom placed at vertices and horizontal velocities at triangles. Because of the dominance of hydrostatic balance, the vertices are aligned in the vertical with the surface vertices making the treatment of the vertical direction similar to that in structured-mesh models. The vertical coordinate is realized using arbitrary Lagrangian–Eulerian (ALE) methodology supporting different choices of model layers. The Finite Element Sea-Ice Model (FESIM, Danilov et al. (2015)) is included in FESOM2 as a set of subroutines. It solves the modified Elastic-Viscous-Plastic (mEVP) dynamical equations allowing to reduce the number of subcycling steps without compromising numerical stability (Kimmritz et al., 2017; Koldunov et al., 2019). The surface mesh partitioning is carried out using the METIS software package. The sea-ice model is run on the same partitioning as the ocean model and is called each step of ocean model. Although this may lead to some unnecessary overhead in sea-ice free regions – especially in global setups – it makes the exchange between the sea-ice and ocean components trivial. Next, we describe the time step structure to the extent necessary for the discussion of scalability.

**Algorithm 1** Time step algorithm for FESOM2.

1: **for all** time steps **do**

2:     Sea-ice update:

3:         Read atmospheric forcing

4:         Ocean2Ice (fields)

5:         Sea-ice dynamics using mEVP approximation: (A10), (A7)–(A9)

6:         Sea-ice advection and thermodynamics

7:         Ice2Ocean (heat and freshwater fluxes)

8:     Horizontal velocity predictor:

9:         Contributions from the pressure gradient and the Coriolis force

10:         Momentum advection and viscosity (implicit vertical viscosity)

11:     External mode:

12:         Update the sea surface height (SSH) stiffness matrix

13:         Compute the SSH right-hand-side based on predicted velocity

14:         Solve for new SSH: (A1)

15:     Horizontal velocity corrector

16:     ALE and vertical velocity step

17:     Tracer advection and diffusion

18: **end for**

The real procedure is more complicated and includes several additional pieces intertwined with those mentioned. Computations of pressure anomaly due to change in density are performed simultaneously with computations of the Brunt–Väisaälä frequency and are taken out of the velocity predictor step. Computations of the thermal expansion, saline contraction, isoneutral slope, the Gent–McWilliams bolus velocity (Gent and McWilliams (1990)), and vertical mixing (K-profile parametrization, Large et al. (1994)) appear in different places before they are first needed and are not timed separately in our study.

The description above does not include model input and output (I/O), which can be time consuming if done too frequently and represents a major scalability bottleneck of its own.

From the standpoint of model's parallel scalability, the procedures mentioned above can be split in two classes. The first class includes the essentially two-dimensional computational parts such as the SSH solver (Appendix A1 and Appendix B) and the sea-ice dynamics (Appendix A2). The mEVP sea-ice solver in FESOM2 (Danilov et al. (2015)) commonly carries out about one hundred shorter time steps within a time step of the ocean model each followed by the halo exchange for two-dimensional sea-ice velocities. The total amount of information being exchanged is comparable to just a single three-dimensional halo exchange, yet there are many such exchanges. The SSH solution is obtained by using an iterative solver utilizing pARMS[1],

---

[1]https://www-users.cs.umn.edu/ saad/software/pARMS/

which also involves multiple two-dimensional halo exchanges and – even worse – global communications within the iterative procedure.

The other class includes the remaining routines. Most of them are three-dimensional and involve either no or only one halo exchange per ocean time step. The tracer advection presents the exception: It may need additional communications depending on the order of the approximation and on whether or not the flux-corrected transport (FCT) procedure is applied. However, in such cases, the amount of numerical work also increases accordingly, thus the code scalability should not be influenced in a major way. One more subroutine containing additional halo exchanges is the horizontal viscosity, which also needs some smoothing if the Leith parametrization is selected.

In the configuration used here to study the scalability, the third/fourth order transport algorithm with FCT is used. It involves four 3D halo exchanges per time step. The Leith parametrization is used as well, thus the number of 3D halo exchanges is at its maximum.

FESOM2 uses vertical as a tighter index than the horizontal to re-use the information on horizontal neighborhood along the vertical column. This is also favorable for vertical operations and generally does not lead to cache misses in the horizontal, especially if the 2D mesh is sorted along a space-filling curve. The tracers are stored in a single array tracer($\text{num}_{levels}$, $\text{num}_{2Dvertices}$, $\text{num}_{tracers}$), and advection and diffusion computations are applied to each tracer separately in a loop over the tracers. In this case, each tracer is contiguous in memory. Auxiliary arrays needed for FCT and high-order advection are allocated for one tracer only.

## 2.2 Test cases and HPC systems used in the study

| Name | 2D vertices | Resolution | Rectangular analogue | Vertical layers |
| --- | --- | --- | --- | --- |
| **CORE2** | 127 000 | 60-25 km | 0.7° | 47 |
| **fArc** | 638 000 | 60-4.5 km | 0.3° | 47 |
| **STORM** | 5 600 000 | 10-3 km | 0.1° | 47 |

**Table 1.** Characteristics of the meshes. Rectangular analogues refer to Mercator-type grids with a similar number of wet points.

Simulations are performed on three meshes (Fig. 1, Tab. 1) using 47 non-equidistant vertical $z$-layers. The first is a low resolution CORE2 mesh specially constructed to better represent global circulation in a low resolution setup. It consist of $\sim$0.13M wet surface vertices. Its horizontal resolution is $\sim$25 km north of 50°N as well as around the coast of Antarctic, $\sim$65 km in the Southern Ocean, and up to $\sim$ 35 km in the equatorial belt. There is also a slight increase in resolution near coastal regions, and most of the ocean interior is resolved by elements sized around 1°. The second mesh, referred to as fArc (FESOM Arctic), aims to better represent circulation in the Arctic Ocean while maintaining a relatively low resolution in the rest of the ocean. It is similar to CORE2 over most of the ocean but is refined down to 4.5 km in the Arctic. It contains about 0.6M surface vertices, which is less than the number of wet vertices in a typical Mercator 1/4 ° mesh. The third mesh is the 1/10 ° mesh

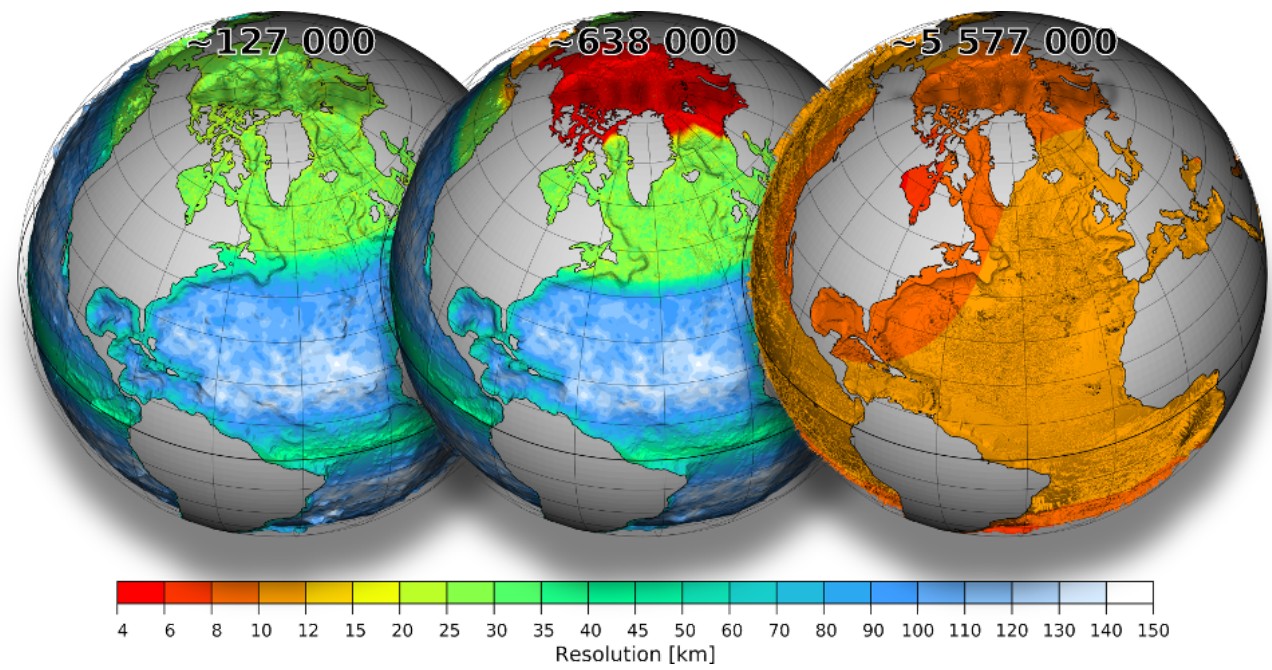

**Figure 1.** Resolution maps for meshes used for our experiments: CORE2 (left), fArc (middle), and STORM (right). Total number of surface vertices for each mesh is displayed at the top.

based on the one used in the STORM project (see, e.g., von Storch et al. (2012)) with surface quads split into two triangles and land quads removed. It contains ca. 5.6M wet surface vertices.

All meshes have 47 unevenly spaced vertical layers. The K-profile parameterization (Large et al., 1994) is used for the vertical mixing, and isoneutral diffusion (Redi, 1982) and the Gent–McWilliams (GM) parameterization (Gent and McWilliams, 1990) are utilized. Note that the GM coefficient is set to 0 when the horizontal grid spacing goes below 25 km. The horizontal advection scheme for tracers uses a combination of third and fourth-order fluxes with flux corrected transport (FCT), for horizontal momentum advection, a second-order flux form is used. The Leith viscosity (Leith, 1968, 1996) is used together with the modified Leith viscosity in combination with weak biharmonic viscosity. The vertical advection scheme for tracers and momentum combines third and fourth order fluxes. Sea-ice dynamics uses the mEVP option (Kimmritz et al., 2017; Koldunov et al., 2019) with 100 subcycles. To ensure model stability for "cold start" experiments with high resolution meshes (fArc and STORM), we use smaller time steps (4 and 2 minutes respectively) compared to the time steps used in production runs (15 and 10 minutes respectively). The time step of 45 minutes is used in all COREII experiments.

Our scaling tests utilize four HPC systems summarized in Tab. 2; also the specific versions of FORTRAN/C compilers and MPI libraries employed in the runs are listed in Tab. 2. For each HPC system used in our runs, the maximum number of (physical) cores per node is used without any hyperthreading (except for the comparisons in Sec. 5.3).

| HPC center | Machine | CPUs | Freq. | Nodes/Cores | Network | Compiler | MPI |
|---|---|---|---|---|---|---|---|
| DKRZ Hamburg | Mistral | Intel Xeon E5-2695 v4 (Broadwell) | 2.1GHz | ca. 3300/36 | FDR InfinBand, 48 Gbit/s | Intel 18.0.1 | OpenMPI 2.0.2 |
| JSC Jülich | JUWELS | Intel Xeon Platinum 8168 (Skylake) | 2.7 GHz | ca. 2500/48 | EDR-Infiniband, 100 Gbit/s | Intel 19.0.1 | ParaStationMPI 5.2.1 |
| AWI Bremerhaven | Ollie | Intel Xeon E5-2697 v4 (Broadwell) | 2.3 GHz | 308/36 | OmniPath, 100 Gbit/s | Intel 19.0.1 | IntelMPI 2018.4 |
| RRZE Erlangen | Emmy | Intel Xeon 2660 v2 (Ivy Bridge) | 2.2 GHz | 560/20 | QDR Infiniband, 40 Gbit/s | Intel 17.0.5 | IntelMPI 2017.5 |

**Table 2.** Overview of the systems used in the scaling experiments.

## 3 Scaling tests

Each simulation in this section starts from a state of rest by reading the mesh, partitioning, initial conditions, and forcing; after that, 1800 time steps are performed; no output is written out during the run. The linear free surface option and the new version of the BiCGStab method for the SSH solver (see Sec. 4.2 for details) are used, and no setup or I/O operations are included in any timings. Whereas the majority of the used configuration settings are either standard for the scalability studies or for FESOM2 production runs, a word of caution with respect to this testing protocol has to be given: Since the model is working with the ocean fields that are not yet fully developed, the time step for high resolution meshes has to be reduced. Our motivation and consequences of using this protocol for scaling experiments are discussed in Appendix B1. Here, we only mention that differences in absolute runtimes of "cold start" with smaller time step vs. "warm start" with larger time step do not affect our general conclusions. Therefore a simple testing protocol can be used, although absolute values for runtimes of individual model components should be considered to be just an approximation.

For the fArc mesh (0.6 M vertices) on Mistral (Fig. 2, top left), we start with a moderate number of cores (144), which is then increased in increments up to 6912. The model shows linear strong scaling in the total runtime up until 1152 cores corresponding to about 500 vertices per core and then starts to slightly deviate from the linear scaling. The two components that clearly account for the bulk of scaling deterioration are the sea-ice and the SSH solver. Note that the rest of the model components continue to scale linearly up to 6912 cores corresponding to ca. 100 vertices per core. The behavior for JUWELS (Fig. 2, top right) is similar although JUWELS is in general somewhat faster.

The scaling of the SSH solver on both machines practically stagnates at some point (after 1152 cores on JUWELS and 1728 cores on Mistral). The sea-ice model scaling is not linear, but the absolute values of the wallclock time still improve almost until the end. The fArc mesh is focused on the Arctic Ocean, therefore the sea-ice computations initially (on 144 cores) occupy about 16% of the total time, while the SSH computations require only 6%. At the greatest number of cores (6912), they occupy already 33% and 28%, respectively. The scaling for the entire model starts to considerably deviate from the linear behavior after SSH and sea-ice calculations become more expensive than the 3D part of the code.

The scaling results for the STORM mesh (5.6M wet vertices) are shown in Fig. 3. On Mistral, our scaling experiments could utilize up to 50,688 cores (by means of a special reservation), while on JUWELS maximum 12,288 cores were available, and all tests were performed using the general queue. The scaling behavior of the STORM mesh is similar to that of fArc: The 3D parts scale practically linearly until about 100 vertices per core, whereas the SSH calculations and the sea-ice model present

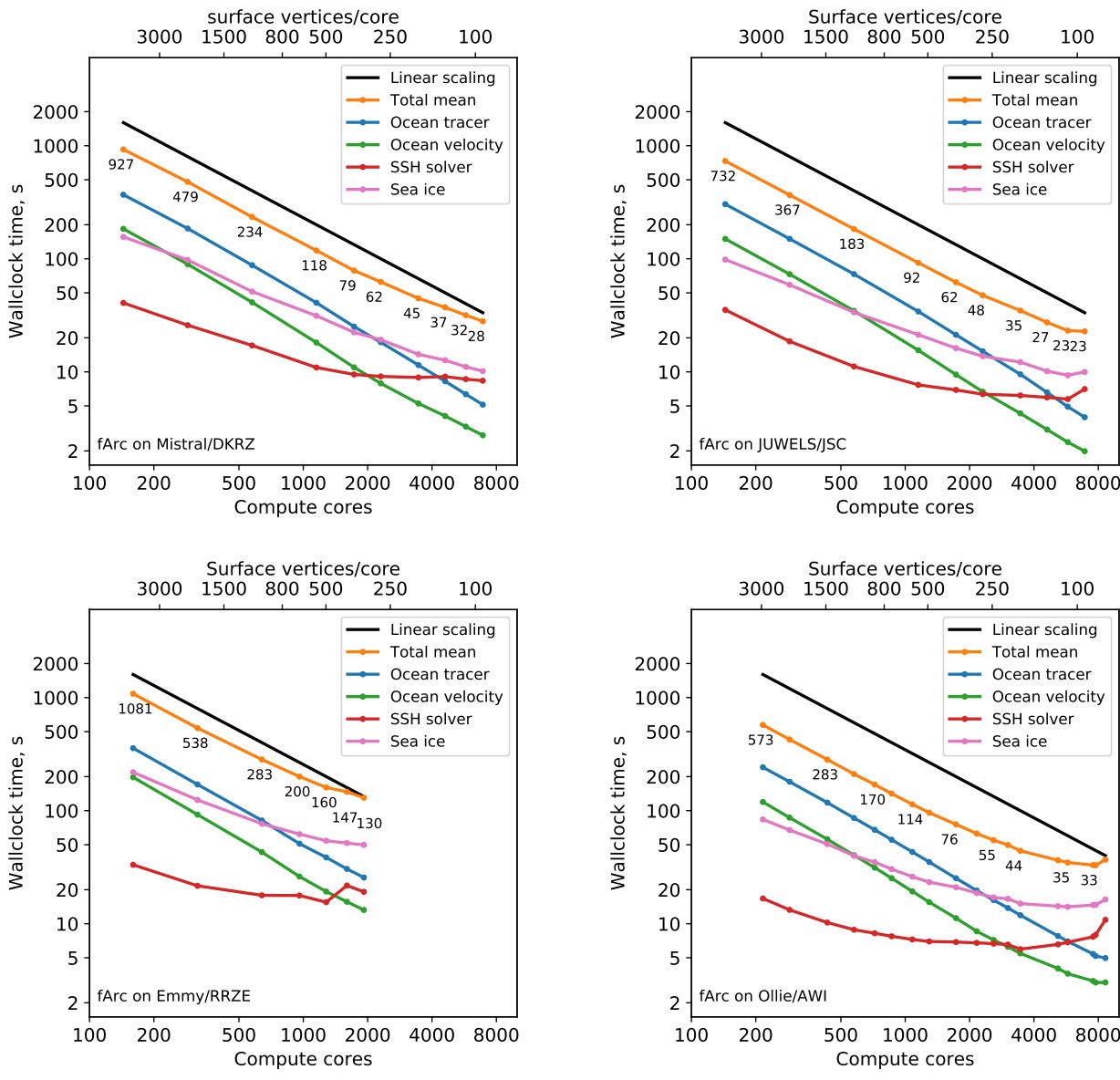

**Figure 2.** Scaling results for the fArc mesh on Mistral (DKRZ Hamburg, top left), JUWELS (JSC Jülich, top right), Emmy (RRZE Erlangen, bottom left) and Ollie (AWI Bremerhaven, bottom right ) compute clusters. The black line indicates linear scaling, the orange line (and the number labels) give the mean total computing time over the parallel partitions.

scalability bottlenecks. After 11,520 cores on Mistral, the SSH runtime stays around 13-14 seconds but, surprisingly, does not get worse. On JUWELS, the SSH stops to scale much earlier, already at 5,760 cores, and oscillates around the runtime value of 15 seconds. The sea-ice on Mistral continues to scale until about 41,472 cores, but the scaling is clearly suboptimal; on JUWELS, the runtime for the sea-ice kernel continues to improve until the maximum number of cores is reached. In the

runtimes for both 2D code parts (sea-ice and SSH), we also clearly notice much stronger oscillations that indicate sensitivity to the state of the communication hard- and software of the corresponding HPC system.

The STORM mesh experiments on Mistral were conducted in two separate series: Up to 18,432 cores in the general queue without any special job scheduler configuration; larger jobs were run using a special reservation for 1,500 nodes. This difference in the experiment setting might be responsible for the sudden oscillations in the timings for the SSH solver and for some other,

perfectly scalable otherwise, routines such as the ocean mixing and computation of the pressure gradient (not shown). After these oscillations, the total time continues to improve until the maximum number of cores – even though it does not go back to the linear scaling path anymore.

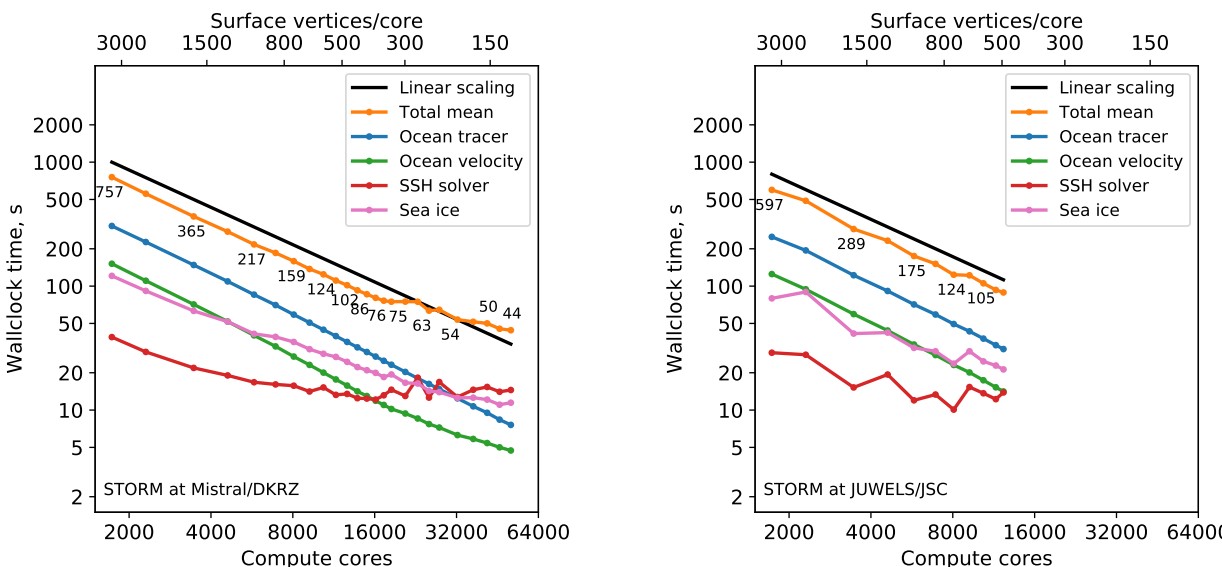

**Figure 3.** Scaling results for the STORM mesh on Mistral (DKRZ Hamburg, left), and JUWELS (JSC Jülich, right) compute clusters. The black line indicates linear scaling, the orange line (and the number labels) give the mean total computing time over the parallel partitions.

We use the smaller CORE2 mesh to demonstrate the model behavior when partitionings with less than 100 vertices per core are used (Fig. 4). The cumulative effect of increasing runtimes of the 2D model parts and decreasing runtimes of the 3D parts

is an almost constant total runtime. This good scaling of the 3D parts of the FESOM2 code suggests that the 3D model parts may be efficiently computed on hardware architectures with low memory per computational core (e.g. GPUs).

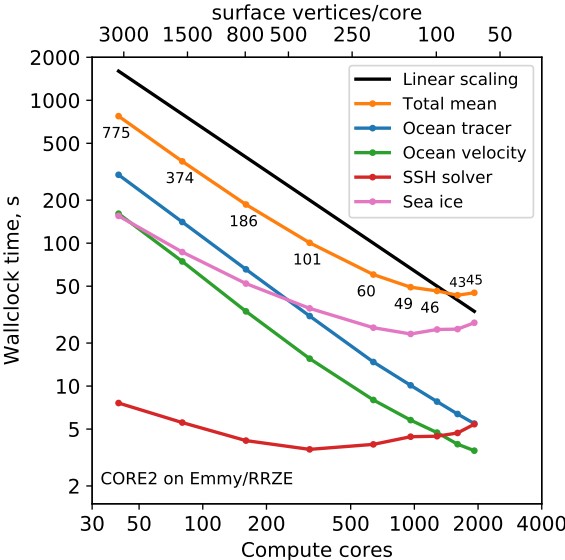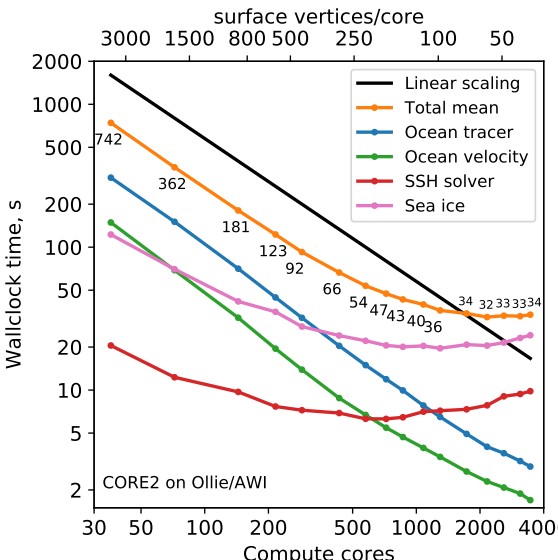

**Figure 4.** Scaling results for the CORE2 mesh on Emmy (RRZE Erlangen) and Ollie (AWI Bremerhaven) compute clusters. The black line indicates linear scaling, the orange line (and the number labels) give the mean total computing time over the parallel partitions.

Runtime values detailing performance of individual model components are presented in Fig. 5. Although obtained in simulations using the CORE2 mesh on 288 cores of Ollie/AWI and simulating 11680 time steps (one model year with 45 minute time step), their relative values are representative for other meshes.

The standard configurations of FESOM2 have 47 unevenly spaced vertical $z$-levels, which is nearly sufficient to resolve the
first baroclinic mode. For setups with high horizontal resolution, it is beneficial to increase the number of vertical levels to better resolve vertical structure of horizontal motions in the ocean (e.g. Stewart et al. (2017)). In order to better understand the effect of increasing the number of vertical layers on model's scaling, we set the number of vertical levels to 71 following recommendations described in Stewart et al. (2017). This compares well with typical numbers of layers used in high resolution model setups. Experiments were performed on Mistral with fArc mesh, and all settings were the same as in the scaling runs
in Sec. 3. Just as expected, with increased number of vertical levels, the scaling improves, while the model become slower (Fig. 6). Deviations from the linear scaling are still observed, but the decline is smaller for 71 vertical layers than for 47. Increases in the number of vertical levels only affect the 3D part of the model by giving each compute core more local work, while the poorly scalable 2D parts are not affected by the change in the vertical layering. The number of communication calls does not change, although there is an increase in the volume transmitted.
To sum it up: Total runtimes for FESOM2 scale linearly until about 400-300 vertices per core and then start to deviate from the linear behavior. The computationally significant 3D parts of the model scale almost linearly to much lower numbers

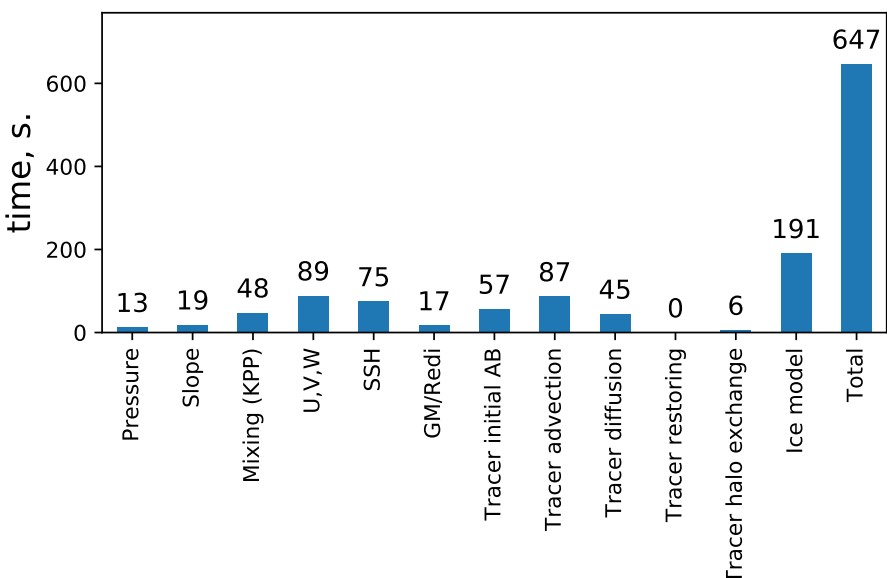

**Figure 5.** Mean wallclock runtimes for different model components. The experiment is run for one year (11680 time steps of 45 minutes) of CORE2 mesh on 288 cores on Ollie/AWI. 'Pressure' includes computations of density and the Brunt–Väissälä frequency, and 'Slope' includes computations of isoneutral slope.

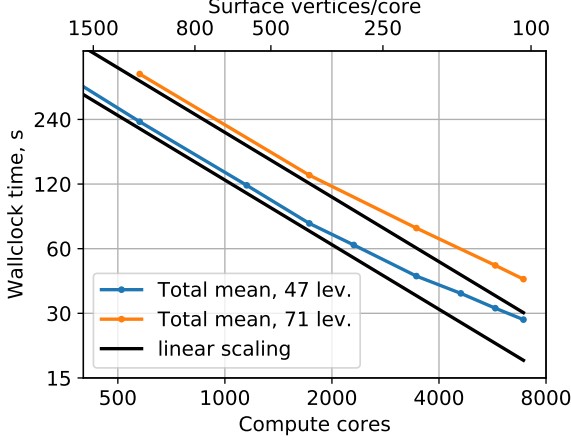

**Figure 6.** The fArc scaling on Mistral (DKRZ Hamburg) with standard 47 layers (blue line) and 71 layers (orange line).

of vertices per core (at least 100), but the main 2D components (SSH and sea-ice) represent the scaling bottlenecks and, by occupying an ever growing fraction of the total runtime, eventually lead to a stagnation and later on to a deterioration in the runtimes with increasing number of compute cores. In practice, for large meshes such as STORM, the limits of scalability are

hardly ever reached using the HPC resources currently available for production runs (e.g., 256 compute nodes on JUWELS and 512 compute nodes on Mistral).

In the following sections, we describe how the above runtimes translate into operational throughputs, discuss several algorithmic and technical enhancements used to reach the current level of scalability, and propose some measures that have the potential to further improve model performance.

## 4 Parallel scaling analysis and techniques to improve parallel scalability

### 4.1 Parallel code analysis using Intel Trace Analyzer

Scaling tests carried out in Sec. 3 show that the major 3D computational parts scale very well, while the scaling of the main 2D kernels, namely of the sea-ice and even more of the SSH solver is substantially worse. To analyze and explain these scaling properties, we visualize the communication patterns of FESOM2 with the Intel Trace Analyzer (ITAC) for a full FESOM2 time step on the CORE2 mesh with 36 (Fig. 7 (top)) and 72 (Fig. 7 (bottom)) MPI tasks on Ollie (AWI Bremerhaven). The different types of computation can be easily distinguished in the ITAC graph:

– First, the sea-ice step with 100 iterations on the 2D mesh performs one halo exchange in every iteration (lines 5, 6 in Algorithm 1). As the CORE2 mesh has no focus in polar regions, most compute cores idle in the MPI_Wait of the halo exchange (light blue in Fig. 7). The zoom in Fig. 7 reveals this heavy load imbalance and also shows the small 2D halo exchanges that are implemented with non-blocking MPI_Irecv and MPI_Isend calls (green) overlapped with a small part of independent calculations (blue) and finalized with MPI_Waitall (light blue).

  The zoom into the sea-ice step (bottom right of Fig. 7, above for 36 cores, below on 72 cores) shows that the computational load scales linearly, but the halo exchange gets more and more expensive.

  At the end of the sea-ice step, the global fresh water balance (line 7 in Algorithm 1) is enforced using MPI_Allreduce (red strip in Fig. 7).

– Second, the horizontal velocity predictor performs 3D computations with a few 3D halo exchanges (lines 9, 10 in Algorithm 1).

– The third part is the 2D solver for the SSH (lines 12–14 in Algorithm 1) dominated by frequent small 2D halo exchanges and global sums (dark vertical strip in Fig. 7). On a small number of cores, the time spent in the solver is low, but it dominates the massively parallel runs. We will go into more detail concerning the SSH solver in Sec. 4.2.

– Finally, the pattern of the computationally intensive 3D part is visible again for the horizontal velocity corrector, the ALE and vertical velocity step, as well as for the tracer advection and diffusion (lines 15–17 in Algorithm 1).

The results shown in Fig. 7, in particular negligible amounts of waiting time, illustrate our claims about highly efficient (in terms of parallel scaling) 3D parts of the FESOM2 code but also explain suboptimal scaling of both main 2D computational

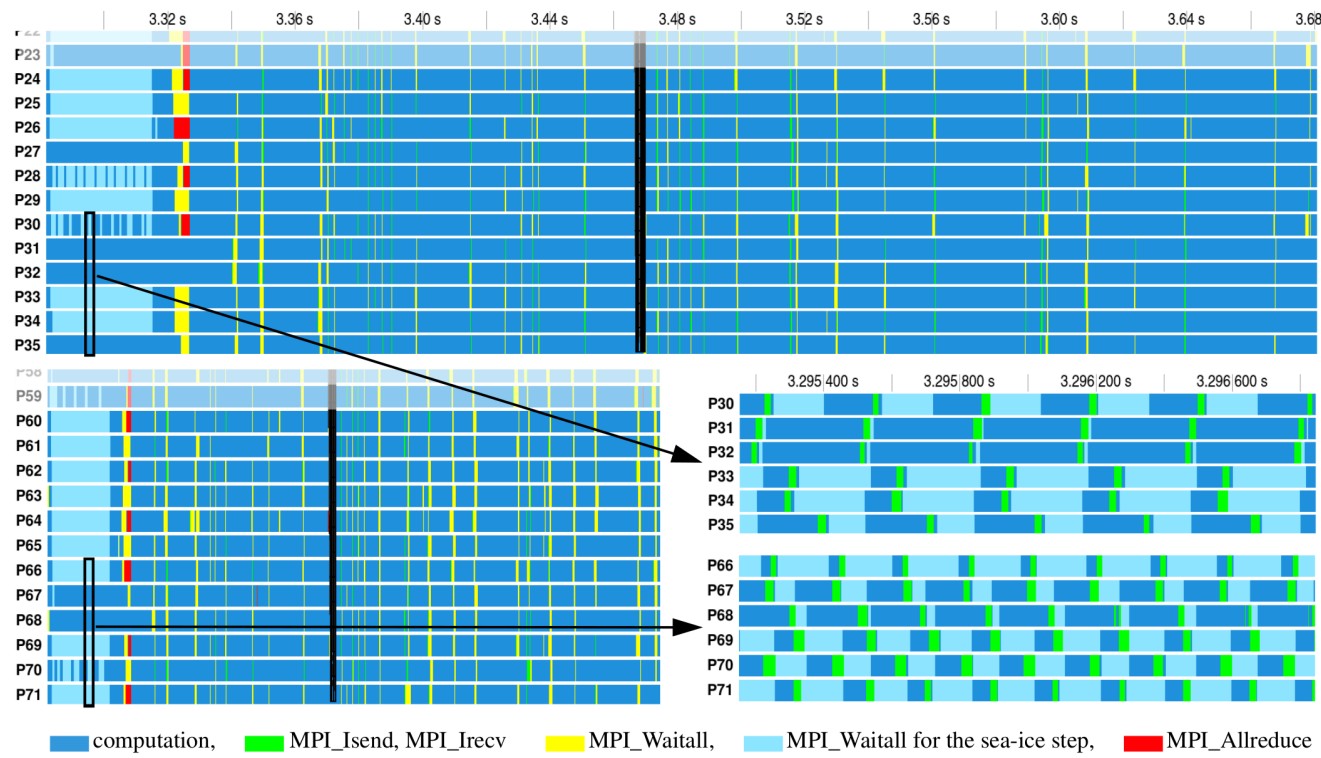

**Figure 7.** One FESOM2 time step on the CORE2 mesh with 36 MPI tasks (above) and 72 tasks (below) visualized in Intel Trace Analyzer. Zoom-ins into iterations of the sea-ice model (bottom right) show the load imbalance between processes with differing amounts of sea-ice per process and illustrate reasons for a suboptimal parallel scaling: With twice the number of cores, the computation time (blue) for parallel partitions fully covered by sea-ice halves; however, the halo exchange time (green) increases.

kernels, the sea-ice and the SSH. Even for rather low numbers of cores, the explicit iterative method for the sea-ice described in Appendix A2 hardly contains enough arithmetic operations to amortize even non-blocking MPI communications involved into 2D halo exchanges. The situation only gets worse when the number of cores is increased and the communication-to-computation ration falls even lower. The analysis of the problems with the SSH scaling and our methods to deal with them are

5 the subject of the next section.

### 4.2 Overlapping computation with communication in the BiCGStab solver for the SSH

For the SSH computation, FESOM2 employs the solver package pARMS augmented by additional Krylov-type solvers and parallel preconditioners (see Fuchs (2013)). For the SSH equation (A1) in both FESOM1.4 and FESOM2, the following settings proved – after extensive testing – to deliver robust results with fast convergence and good parallel efficiency for up to ca. 1000

10 cores (a typical setup for large FESOM runs around 2013):

- BiCGStab Krylov solver (Algorithm 2),

- Parallel preconditioner restrictive additive Schwarz (RAS) with one vertex overlap,

- Local preconditioner incomplete LU-factorization ILU(k) with fixed fill level $k = 3$,

- Solution tolerance $10^{-8}$.

Our recent scaling tests revealed that the SSH solver with the above settings represents a major bottleneck for the next level of scaling: Up to 10,000 compute cores and beyond. Since the iterative SSH solver is very similar in its parallel communication patterns to the sea-ice step with frequent 2D halo exchanges and little computational work in-between, a substantially worse scaling of the SSH step turns out to be caused by the additional global communications. Fig. 9 (top) shows one BiCGStab iteration with three MPI_Allreduce calls (see also Appendix B): The first two calls performed after the matrix-vector product

and the preconditioner step cause increased waiting times. In addition to the blocking behavior, the MPI_Allreduce itself becomes expensive on higher core counts. Another limiting factor is the convergence of the RAS preconditioner, whose efficiency depends on the sizes of parallel partitions (see Smith et al. (2004)), and, as a consequence, the iteration count increases slowly with the number of compute cores.

    After searching for alternatives, we implemented the pipelined BiCGStab algorithm (Algorithm 3) proposed in Cools and

Vanroose (2017) that resulted in an improved scaling behavior (see Fig. 8) especially for high core counts. As opposed to the classical BiCGStab algorithm, the pipelined version replaces the three blocking MPI_Allreduce calls by two non-blocking communications overlapped with the computations of the matrix-vector-product and the preconditioner (see Fig. 9 (bottom)). Since the pipelined version involves more arithmetic operations for vector updates, it may be inferior to the original BiCGStab method on lower core counts. Also differences in the HPC hardware play a substantial role in the comparative performance of

both algorithms as clearly illustrated in Fig. 8.

## 4.3   Speeding up the sea-ice model

The equations of sea-ice dynamics with traditional viscous-plastic rheology and elliptic yield curve (A2)–(A4) are very stiff and would require time steps on the level of a fraction of a second if computed explicitly (see, e.g., Kimmritz et al. (2017) for a brief summary). For this reason, they are solved either with an iterative solver or explicitly using pseudo-elasticity to reduce the

limitations on short time step as proposed by Hunke and Dukowicz (1997). The latter approach is called elastic-viscous-plastic (EVP) and requires $N_{\mathrm{EVP}} \geq 100$ substeps within the external step of the sea-ice model. Other components of the sea-ice model such as the advection or the computation of thermodynamic sources and sinks are advanced at the time step of the ocean model and are much less demanding. Since the EVP approach is explicit, it has to satisfy stability limitations – as discussed by many authors beginning from the original paper by Hunke and Dukowicz (1997). It turns out that, as the mesh resolution is refined,

$N_{\mathrm{EVP}}$ must be increased to maintain stability and becomes prohibitively large. The violation of stability leads to noise in the derivatives for the sea-ice velocity. Although this noise may stay unnoticed by users, it affects dynamics.

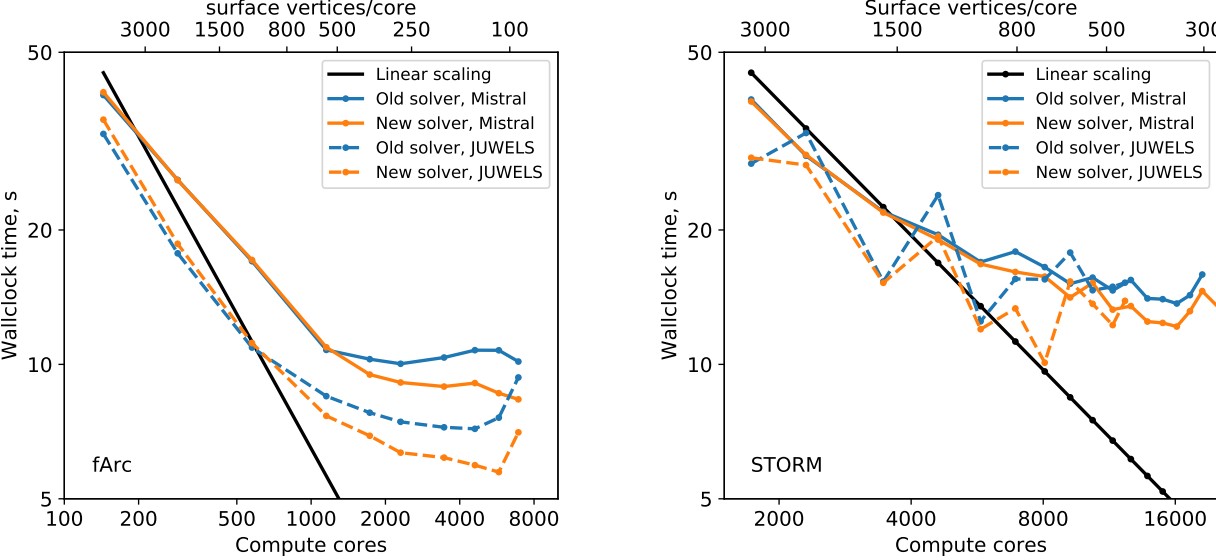

**Figure 8.** Scaling results for old (Algorithm 2) and new (Algorithm 3) SSH solvers for fArc (left) and STORM (right) meshes on Mistral (DKRZ Hamburg) and JUWELS (JSC Jülich) compute clusters.

An essential measure taken in FESOM2 to improve scalability is to use the modified EVP approach (mEVP) as described in Danilov et al. (2015). As opposed to the traditional EVP approach, the mEVP approach, based on a suggestion by Bouillon et al. (2013), splits the issues of numerical stability and convergence. This solver is always stable, and the number of the substeps $N_{\mathrm{EVP}}$ determines its convergence to the viscous-plastic rheology. In practice, it turns out that mEVP produces practically acceptable solutions in the regime with relatively small $N_{\mathrm{EVP}}$ that only corresponds to an initial error reduction. $N_{\mathrm{EVP}}$ is found experimentally by starting simulations from the values greater than the stability parameters $\alpha$ and $\beta$ in (A7)–(A10) and reducing them to minimum values that lead to results practically indistinguishable from simulations performed with a large $N_{\mathrm{EVP}}$ – as explained in Koldunov et al. (2019). We use $N_{\mathrm{EVP}} = 100$ for both fArc and STORM meshes. These values are already much lower than the values needed for the standard EVP, and, in this sense, the sea-ice model is already optimal. The sea-ice model uses only local communications, but their number is proportional to $N_{\mathrm{EVP}}$ per external time step.

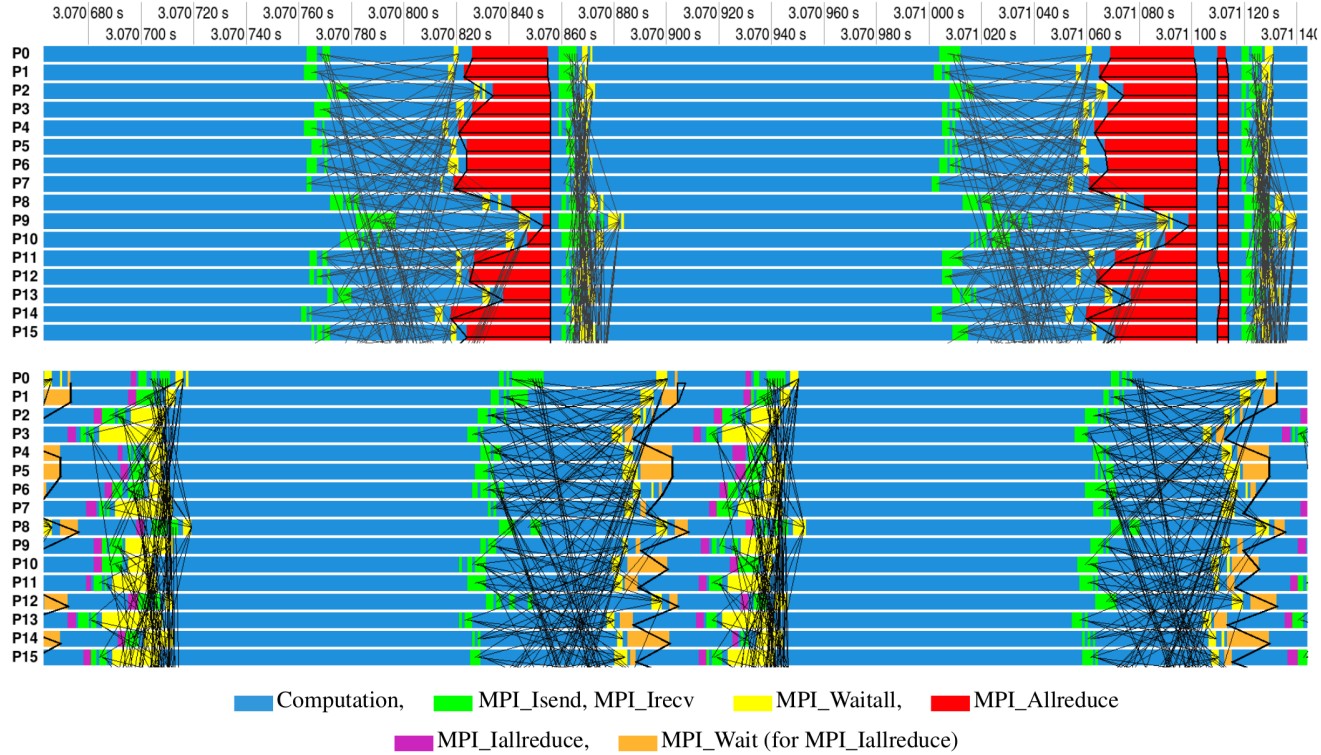

**Figure 9.** Above: One iteration of the classical BiCGStab scheme (Algorithm 2) in the Intel Trace Analyzer. Observe the blocking due to computing global sums in scalar products via MPI_Allreduce. In addition, communication is needed for halo exchanges that often overlap with independent computations. Below: One iteration of the pipelined BiCGStab scheme (Algorithm 3). The global sums are now initiated by a non-blocking call to MPI_Iallreduce and finished by MPI_Wait. In between, major computations such as a matrix-vector product and a preconditioning step are performed also including halo exchanges.

## 5 Further tests

### 5.1 Testing throughput in an operational configuration

The scaling tests are by design performed in a somewhat idealized setting that allows us to minimize the effect of factors not related to pure computation (see also Appendix B1). However, in the operational setting, aspects such as the I/O frequency, the dependence of the number of iterations in the SSH solver on the model state and model diagnostics might require additional computations. In Fig 10, the throughput estimates in simulated years per day (SYPD) are plotted for selected scaling results from Sec. 3 against estimates based on one-year computations with I/O on Mistral/DKRZ conducted in an operational configuration. The one-year experiments were started using the restart files produced by one year of model spin-up. After one year of simulation time, the model dynamics are usually well developed, and the time step sizes as well as the velocities have values typical for production runs. The restart files were written out once per year, and the standard model output was also switched

on, namely monthly 3D fields of temperature, salinity, three velocity components, and monthly 2D fields of sea surface height, sea surface temperature, sea surface salinity, as well as the sea-ice fields: concentration, thickness, and two components of ice velocity. Estimates obtained from scaling experiments are calculated with the assumption of using the same time step as in one-year runs (15 minutes for fArc and 10 minutes for STORM). Estimates obtained for one-year runs are calculated without accounting for the initial model setup (reading mesh, restart files, etc.) needed only once per run.

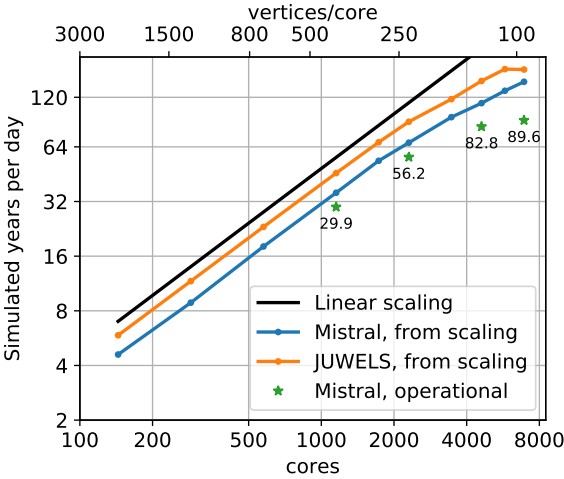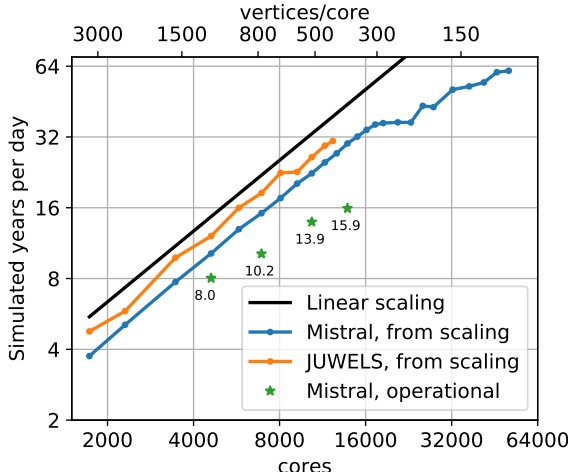

**Figure 10.** Model throughput in simulated years per day (SYPD) estimated from scaling experiments and measured in one-year experiment with I/O for fArc (left) and STORM (right).

The SYPD estimates based on one-year experiments are substantially lower than those obtained from the scaling results; we attribute this slowdown of the model in an operational regime to the I/O overhead and the SSH solver. For fArc at 1152 cores, I/O operations (mainly writing the recurring output files and the restart at the end of the year) use ca. 5% of the time, while, at 6912 cores, it becomes 17%. Even worse is the situation for STORM with I/O at already 25% of computing time at 4608 cores and growing to 65% at 13824 cores. Note that the amount of the I/O in our one-year experiment is rather moderate (limited number of monthly fields). In high resolution (eddying) regimes, diagnostics based on the mean effects of eddies require much higher output frequencies. The I/O in FESOM2 is still organized in a serial way with fields from all computational cores collected on one core and written to disc. While sufficient on small meshes, this methodology is obviously a bottleneck for simulations using large meshes on large parallel partitions. It will be changed in the nearest future in favor of parallel I/O solutions.

Also the fraction of the total runtime (excluding I/O) spent in the SSH solver is greater for one-year simulations than for scaling experiments due to larger time step. The increase is due to poorer conditioning of the SSH matrix and consequent

increase in the number of SSH solver iterations. More details on the consequences of this issue for our scaling experiments are presented in Appendix B1.

The real model performance depends on many factors, including I/O, state of the machine, number of vertical levels, time step and other details of the model configuration. Therefore any timings produced in idealized settings must be taken with a grain of salt when model throughputs lie in the focus of a discussion. It is even harder to compare throughput of different models to each other, especially when not all details are available. Nonetheless, an attempt to make such a comparison is presented in Table 3, which should be treated as a qualitative estimate.

Nevertheless an attempt to account for differences in model configurations is made by introducing $c_{SYPD}$ constants. The relationship between resulting SYPD, basic model characteristics, and computer resources can be expressed as:

$$SYPD = c_{SYPD}\frac{\Delta t * N_{core}}{N_{3D}},$$

so the SYPD is directly proportional to model time step ($\Delta t$) and number of computational cores ($N_{core}$), while inversely proportional to the number of degrees of freedom ($N_{3D}$). The larger the proportionality constant $c_{SYPD}$ the more computationally efficient is the model. In Table 3, we list $c_{SYPD}$ for 3D (vertices*number of layers) and 2D degrees of freedom (vertices) since 2D model parts solvers take up a significant fraction of computational time.

This comparison shows that, once the differences in the number of model levels, computational resources, time steps, and mesh size are accounted for, the FESOM2 throughput is easily on a par with that of well-established structured models confirming our claim that unstructured-mesh models can be about as fast as conventional structured-mesh models and thus represent an efficient tool for ocean and climate studies. Considering overall model scalability, there is still room for improvement, especially for greater numbers of computational cores, by transitioning to a parallel I/O.

| Model/mesh | Resolution | Vertices (ocean) | Cores | Time step, s | Levels | SYPD | $c_{SYPD}$3D | $c_{SYPD}$2D | Citation |
|---|---|---|---|---|---|---|---|---|---|
| POP | 1/10° | 5.8M | 16875 | 173 | 60 | 10.5 | 1252 | 20 | Huang et al. (2016) |
| ACCESS-OM2-01 | 1/10° | 5.8M | 6138 | 450 | 75 | 1.2 | 188 | 3 | Kiss et al. (2019) |
| FESOM2/STORM | 1/10° | 5.6M | 13828 | 600 | 47 | 15.9 | 505 | 11 | |
| NEMO/ORCA25 | 1/4° | 0.9M | 2048 | 3600 | 75 | 5-10 | 92 | 1 | Prims et al. (2018) |
| MOM5.1/CM2.5 | 1/4° | 0.9M | 960 | 1800 | 50 | 11 | 286 | 6 | Ward and Zhang (2015) |
| MOM6 | 1/4° | 0.9M | 1920 | 1200 | 75 | 8.9 | 260 | 3 | Ward (2016) |
| ACCESS-OM2-025 | 1/4° | 0.8M | 1816 | 1800 | 50 | 9 | 110 | 5 | Kiss et al. (2019) |
| FESOM2/fArc | 1/3° | 0.6M | 2304 | 900 | 47 | 56.2 | 764 | 16 | |
| ACCESS-OM2 | 1° | 0.065M | 240 | 5400 | 50 | 63 | 158 | 3 | Kiss et al. (2019) |
| FESOM2/CORE2 | 1° | 0.13M | 288 | 2700 | 47 | 120 | 921 | 20 | |

**Table 3.** Throughput of different models at selected configurations comparable to FESOM2 STORM, fArc and CORE2 meshes. Vertices are 2D degrees of freedom only in the ocean. It is not possible to express the resolution of FESOM2 configurations in a single number, therefore we list resolutions that regular Mercator grid would have with a similar number of ocean degrees of freedom. Data for FESOM2 configurations are taken from one year simulations with I/O.

## 5.2 Hierarchic mesh partitioning

The standard way of performing mesh partitioning in FESOM2 relies on METIS (Karypis and Kumar (1998)) graph partitioning package (currently Version 5.1.0) and utilizes the dual-weighted load balancing criterion based on the number of 2D and 3D mesh vertices. This approach works well for moderate numbers of parallel partitions but tends to produce some undesirable artifacts for large numbers of partitions: Isolated vertices, partitions containing vertex groups widely separated in the computational domain, non-contiguous partitions, etc.

As a simple but elegant way to remedy some of these deficiencies, a backward compatible wrapper for METIS has been implemented that allows to perform the mesh partitioning in a hierarchic fashion (see Fig. 11 (left)). The procedure starts from the coarsest level, e.g., producing a partition per networking switch. Then METIS is called recursively for each partition on the current level until the lowest level (usually that of a single core) has been reached. Since the performance of METIS for small numbers of partitions is usually excellent, this approach guarantees that each coarse partition only contains contiguous vertices thus potentially improving the mapping of the computational mesh onto the topology of the compute cluster. This methodology is certainly not entirely new (see, e.g., the method called nested partitioning in Sundar and Ghattas (2015)); however, neither implementations of this idea as our simple METIS wrapper nor applications of this technique to ocean modeling could be found in the literature.

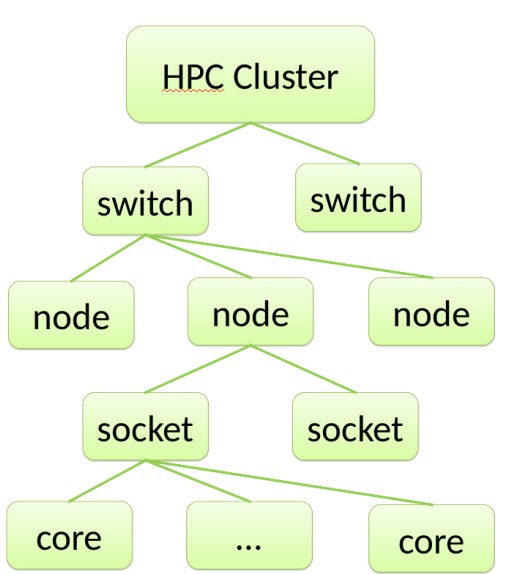 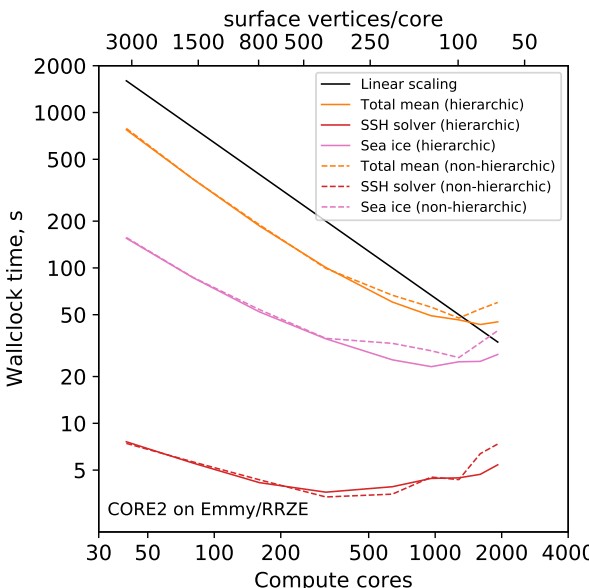

**Figure 11.** Schematics of an HPC cluster for purposes of hierarchic mesh partitioning (left) and a comparison of the total, SSH, and sea-ice scaling for hierarchic (continuous lines) and non-hierarchic (dashed lines) partitionings for the Core2 mesh on Emmy (RRZE Erlangen, right).

The simulation runs with the Core2 mesh on Emmy (Fig. 11 (right)) appear to show a small edge for the hierarchic partitioning in cases if the amount of work per partition becomes small; however, runs on HPC systems with higher-end network hardware such as Ollie or JUWELS show no discernible differences to non-hierarchic partitioning. We attribute this discrepancy to the ability of new communication hardware with its superior efficiency and lower latency to hide deficiencies in the mesh partitioning, whereas older network hardware on Emmy is more sensitive to the placement of grid vertices on parallel partitions. A similar situation – however on a much higher level – may also arise in the future if the communication efficiency in the future HPC systems does not keep pace with the computational performance increases (e.g., rises in the number of compute cores) of compute nodes.

In addition to improving the mapping of the partitioning topology to the topology of the HPC system, the hierarchic partitioning capability is intended to provide a simple interface for placing poorly scalable computational kernels (e.g., I/O or sea-ice) at a certain mesh hierarchy level with a natural way to supply clearly defined relationships between partitions on different levels. By exploiting this feature, we hope to be able to improve computational efficiency of kernels currently bound by the I/O or communication bandwidth and to also provide interfaces to hardware accelerators such as GPUs or FPGAs.

### 5.3 Improving the network performance by underpopulating compute nodes

A simple but often effective way to improve the network performance on the current-generation network hardware such as Intel Omnipath is to leave one core per node off the regular compute job assignment. Indeed, our experiments on Ollie and Mistral (not shown) demonstrated little difference between fully assigned and underpopulated nodes (or sometimes even a slightly better SSH timings for underpopulated nodes). In this way, in addition to networking duties, other loads (e.g. I/O) currently distributed over all compute cores can be offloaded to these idle nodes.

## 6 Discussion of results and strategies for improvements in model scalability

### 6.1 SSH strategy

Iterative solvers based on preconditioned conjugate gradient method or other Krylov subspace methods need global communications, which is the reason behind many present-day models opting for the split-explicit subcycling to explicitly resolve surface gravity waves. The advantage of using a linear solver for the SSH is its algorithmic simplicity and, in the case of FESOM, enhanced stability on arbitrary meshes (less control on mesh quality and smoothness is needed). Many possible subcycling algorithms have been discussed in the literature (see, e.g., an analysis in Shchepetkin and McWilliams (2005)). We plan to include the subcycling option in FESOM2 in the future, but, judged from the scalability of purely explicit sea-ice solver, only partial improvements can be expected. Indeed, the scalability of an explicit sea-ice vs. an implicit SSH solver is machine dependent (see, e.g., STORM results for the sea-ice and the SSH on Mistral and JUWELS in Fig. 3) and is also never optimal. However, even small improvements in scaling, at least on some machines, might be sufficient to make practicable partitions as small as 200 surface vertices per core. We will report on our results in due time.

Alternatively, one may follow the suggestion of Huang et al. (2016) and switch to a solver based on a Chebyshev method. The method of Huang et al. (2016) is based on an algorithm without global sums and allows one to better overlap computation with communication. A caveat here is that unstructured meshes used by FESOM2 may cause the SSH system matrix to be more poorly conditioned than in Huang et al. (2016), since mesh elements can differ by a factor of 100 or greater. It remains to be seen whether this method is able to compete with the split-explicit approach. Another simple optimization avenue presents itself via renumbering of mesh vertices: In the solver package pARMS, the arrays are permuted to have the inner vertices first, followed by the vertices that have to be send to the neighbouring partitions, and, last, the vertices to be received; this allows to exchange the halo and meanwhile perform computations on the inner vertices. The same strategy can be easily extended to the FESOM2 sea-ice model and to all other halo exchanges. The optimal data structure for unstructured meshes is still under discussion (MacDonald et al., 2011).

## 6.2 Sea-ice model strategy

The sea-ice model is linked less directly to the ocean model than the internal mode, and the simplest choice to reduce its cost would be to call it every second or every third time step of the ocean model. Eddy-resolving codes are usually run with a time step smaller than 10 min, and the sea-ice state does not change much in this time interval. However, increasing an external sea-ice step has implications for stability. Theoretically, running the mEVP with a doubled external time step would require a $\sqrt{2}$ increase in stability parameter with corresponding changes in $N_{\mathrm{EVP}}$. Thus, in the end, one would expect only $\sqrt{2}$ reduction in the time spent on sea-ice computations, yet this has to be carefully tested. One more caveat is the floating (as opposed to levitating) sea-ice option, which might require further adjustments.

A more dramatic change would be to run the sea-ice synchronously with the ocean on a separate set of cores using the same ocean mesh partitioning or partitions obtained by hierarchically combining several ocean partitions to simplify the ocean-sea-ice communication. On global meshes, where the domain covered with sea-ice occupies only a fraction of the total area, the sea-ice model would require only a small number of additional cores. In this approach, one would choose the number of cores for the sea-ice tuned to achieve the runtimes not exceeding those of the ocean model. Clearly, the scalability of the sea-ice model still remains a bottleneck; however, the overall scalability would be substantially improved by running the components with poor scalability in parallel instead of one after another. In addition, this technique can optimize the utilization of computational resources, since the cores currently being idle due to the communication-bound sea-ice model can be used for the well-scalable ocean model. The drawback of this approach is a more complicated code design and the need of a smarter parallel load balancing algorithm. It can be simplified by using couplers, but this may come with its own issues regarding scalability.

## 6.3 Need for more efficient 2D computations

We saw above that, independent of the solution method (iterative solver or subcycling), the two-dimensional parts of the ocean code present a challenge on the road to a better parallel scalability. While the measures briefly discussed above can be helpful, we do not expect that they can deliver optimal scalability down to less than 100 surface vertices per core seen for the 3D part

of the code. The open question is whether new hardware architectures can offer better options. The problem with the external mode and sea-ice solvers lies in too frequent but short communications. On present architectures, a possible strategy to reduce the number of communications in the SSH and sea-ice parts of the code is to make halos wider.

In this regard the concept of dwarfs, also known as Berkeley dwarfs (see Asanović et al. (2006)), further developed and customized for weather and climate applications in Lawrence et al. (2018), might be useful. The main idea behind this concept is to capture some essential characteristics of a model part or a parametrization in a stand-alone piece of software that can be run, analyzed, ported, optimized on its own. In the HPC-speak, a dwarf is a computational kernel enhanced by the ability to run without the remainder of the model and whose performance is critical to that of the whole model. By utilizing dwarfs extracted from the problematic 2D kernels, a number of numerical, algorithmic, and computational methodologies can be tested and optimized in the search for better solutions. In the framework of the currently funded EU project ESCAPE, dwarfs for several weather and climate models are being actively developed (http://www.hpc-escape2.eu/, last access: 19 June 2019).

## 7   Conclusions

We systematically explored the scalability of FESOM2 for a set a meshes (0.13M, 0.64M, and 5.6M surface vertices, 47 layers) on several HPC systems (up to 50 000 compute cores) and found a nearly linear scalability of FESOM2 code to the limit of 300–400 surface mesh vertices per core. Our numerical experiments indicate that the 3D parts of the code scale almost optimally down to 100 surface mesh vertices per core. Similar to other large-scale ocean circulation models, the factor limiting the scalability of FESOM2 is its two-dimensional part, represented by the solver for the sea surface height and the sea-ice model. Since the sea-ice model uses explicit time stepping and local communications, it generally demonstrates better scalability than the solver part; however, this behavior is machine dependent and never reaches the optimal levels of scaling.

Our results also allow us to conclude that the technology of mesh partitioning relying on the connectivity pattern and based on METIS software package in the case of FESOM2 works well on large meshes and down to very small partitions. While the hierarchic mesh partitioning proposed here does not generally improve the scalability, it lays the base for future model improvements, especially concerning the sea-ice model, I/O, and the SSH computation. This methodology also can become useful for unstructured model data parallel post-processing using approaches presented in Rocklin (2015) and Hoyer and Hamman (2017).

The analysis of scalability was complemented by the analysis of model throughput. We show that the FESOM2 not only scales well but also reaches throughputs that are very comparable to those of structured mesh models given the same computational resources and similar meshes. They can be even higher because of a better scalability for high-resolution configurations. Since new unstructured-mesh large-scale ocean model developments (FESOM2, MPAS, ICON) are all based on the finite-volume methodology, similar behavior can be expected in all these cases. In summary, therefore, we see the following statement as the main message to the oceanographic community: The present-day unstructured-mesh global ocean circulation models have reached a level of maturity, where their computational speed is no longer a limiting factor.

While the result that the external mode and the sea-ice model are currently limiting the parallel scalability (with further practical complications arising from the implementation of the I/O) did not come as a surprise, the lesson learned from the analysis presented in this study is the extent of the problem. Given that computational resources available for most current long-term simulations are in many cases still limited to 5000-10000 or less cores, the parallel scalability is only beginning to emerge as a major issue on large (1/10° or finer) meshes. The current CPU architectures appear to be well-suited for nearly all 3D computational parts of FESOM2, thus the potential for improvement seems to lie in the direction of improved memory bandwidth, lower communication latency and more efficient file systems – this can be then used as an indication when choosing 'optimal' hardware. Suboptimal scaling of the sea-ice combined with a sequential arrangement of sea-ice and ocean steps result in an inefficient utilization of computational resources and indicate clear directions for improvement. This, together with a better scalable parallel I/O are the directions for future model code development to enable high-resolution climate simulations with reasonable throughputs.

In order to better prepare ocean modelling community to challenges that come with extreme-scale computing, more efforts should be dedicated to in depth scalability analysis of computational cores in different ocean models and careful intercomparison of ocean model throughputs. This only can be done in coordinated manner similar, for example, to Balaji et al. (2017).

*Code availability.* The current version of FESOM2 is available from the public GitHub repository at https://github.com/FESOM/fesom2 under the GNU General Public License (GPLv2.0). The exact version of the model used to produce the results used in this paper is archived on Zenodo *Danilov et al. (2018)*

*Data availability.* The spreadsheets with results of the measurements are available in .ods and .xlsx formats at Zenodo (Koldunov et al., 2018).

## Appendix A:  SSH and sea-ice computations

Model equations of FESOM2 and their numerical discretization are described in Danilov et al. (2017) and Danilov et al. (2015) for the ocean and the sea-ice parts, respectively. For convenience, we briefly discuss the sea surface height equation and equations solved for the sea-ice dynamics, which are the main stumbling blocks on the way to better parallel scalability.

### A1    Sea surface height equation

The propagation speed of long surface gravity waves is $c = (gH)^{1/2}$, where $g$ is the acceleration due to gravity and $H$ the ocean thickness. Since $c$ may exceed 200 m/s over deep regions, it limits the time step of the ocean model. To circumvent this difficulty, two approaches are used in ocean models. The first one solves the equation for the SSH implicitly or semi-implicitly, whereby the SSH is stepped with the ocean time step at very large Courant numbers $C = c\Delta t/\Delta x$ with respect to the speed of

surface gravity waves. Here $\Delta t$ is the ocean time step and $\Delta x$ is the characteristic size of mesh elements. It implies the use of a linear (usually iterative) solver. The second methodology resorts to the separation of the barotropic dynamics (governed by the SSH and also called the external mode) and stepping it with small time step $\delta t$, generally chosen as a fraction of the ocean time step $\Delta t$. FESOM2 uses the first approach solving (see Eq. (18) in Danilov et al. (2017)):

$$\frac{1}{\Delta t}(\eta^{n+1} - \eta^n) - \alpha\theta g\Delta t\nabla \cdot \int^{\overline{h}^{n+1/2}} \nabla(\eta^{n+1} - \eta^n)dz = R_\eta, \tag{A1}$$

where $\eta^n$ is the elevation on time step $n$, $R_\eta$ the right hand side, $\overline{h}^{n+1/2}$ the position of surface at half time step, and $\alpha$ and $\theta$ are the parameters varying between 1/2 and 1. This equation is obtained by combining the momentum and the elevation equations in a predictor–corrector method. The case $\alpha = \theta = 1$ corresponds to a fully implicit free surface. The limit $\alpha = \theta = 1/2$ corresponds to the semi-implicit Crank–Nicholson method. In scaling experiments reported here, FESOM2 was run with a fully implicit free surface. Although the semi-implicit option reduces the conditioning number of the matrix that corresponds to the left hand side operator, it leaves spurious waves in solution, and there is very little benefit in using it. The left hand side matrix operator is assembled on every time time step due to the variable thickness, except for the linear free surface option, where these changes are neglected. This procedure does not require halo exchange and does not limit scalability. The same concerns the computation of the right hand side $R_\eta$. The main computational burden is related to the iterative solution of (A1), when the number of mesh vertices per core is going down.

## A2   Sea-ice dynamics equations

Advection of sea-ice fields and computations of thermodynamic sources and sinks require only a small computational effort, and nearly all time in the sea-ice routines is spent on solving sea-ice dynamical equations. These equations comprise three tracer transport equations for the sea-ice concentration $a$, sea-ice mean thickness (volume per unit area) $h_{ice}$, and snow mean thickness $h_s$

$$\partial_t a + \nabla \cdot (\boldsymbol{u}a) = S_a, \ \partial_t h_{ice} + \nabla \cdot (\boldsymbol{u}h_{ice}) = S_{ice}, \ \partial_t h_s + \nabla \cdot (\boldsymbol{u}h_s) = S_s \tag{A2}$$

with $S_a$, $S_{ice}$ the sources related to sea sea-ice melting and freezing, and $S_s$ the sources due to snow precipitation and melting; as well as the momentum equation

$$m\partial_t \boldsymbol{u}_{ice} = a\boldsymbol{\tau} - mf\boldsymbol{k} \times \boldsymbol{u}_{ice} - aC_d\rho_w(\boldsymbol{u}_{ice} - \boldsymbol{u})|\boldsymbol{u}_{ice} - \boldsymbol{u}| + \nabla \cdot \boldsymbol{\sigma} - mg\nabla\eta, \tag{A3}$$

where $\boldsymbol{u}_{ice} = (u_{ice}, v_{ice})$ is the sea-ice velocity, $m = \rho_{ice}h_{ice} + \rho_s h_s$ the combined mass of sea-ice and snow per unit area, $\rho_{ice}$ and $\rho_s$ the densities of sea-ice and snow, respectively, $f$ the Coriolis coefficient, $\boldsymbol{k}$ the unit vertical vector, $C_d$ the ice-ocean drag coefficient, $\rho_w$ the water density, $\boldsymbol{u}$ the ocean velocity at the ocean-ice interface, $\boldsymbol{\tau}$ the wind stress at the sea-ice surface, and $\boldsymbol{\sigma} = \{\sigma_{ij}\}$ the internal ice stress tensor computed using the viscous-plastic (VP) rheology (Hibler (1979))

$$\sigma_{ij} = 2\eta(\dot{\epsilon}_{ij} - (1/2)\delta_{ij}\dot{\epsilon}_{kk}) + \zeta\delta_{ij}\dot{\epsilon}_{kk} - (1/2)\delta_{ij}P, \tag{A4}$$

where $\delta_{ij}$ is the usual Kronecker delta, $\eta$, $\zeta$ are the moduli ('viscosities'), and $P$ is the sea-ice strength. The strain rate tensor $\dot{\epsilon}$ is defined as

$$\dot{\epsilon}_{11} = \partial u/\partial x, \quad \dot{\epsilon}_{22} = \partial v/\partial y, \quad \dot{\epsilon}_{12} = \dot{\epsilon}_{21}\dot{\epsilon}_{21} = (1/2)(\partial u/\partial y + \partial v/\partial x).$$

The sea-ice strength $P$ and viscosities are computed as

$$P = P_0, \quad \zeta = (P_0/2)/(\Delta + \Delta_{min}), \quad \eta = \zeta/e^2, \tag{A5}$$

where

$$P_0 = h_{ice}p^* \exp{-C(1-a)}, \quad \Delta^2 = (\dot{\epsilon}_{11}^2 + \dot{\epsilon}_{22}^2)(1 + 1/e^2) + 4\dot{\epsilon}_{12}^2/e^2 + 2\dot{\epsilon}_{11}\dot{\epsilon}_{22}(1 - 1/e^2), \tag{A6}$$

$e = 2$ and $C = 20$; as the default values in FESOM2, we set $\Delta_{min} = 2 \cdot 10^{-9}$ s$^{-1}$ and $p^* = 27500$ N/m$^2$.

The main difficulty of solving (A3) with (A4) is that typical values of viscosities are very large leading to a very stiff and highly nonlinear system. In most cases it is solved by augmenting (A4) with pseudo-elasticity, which results in a elastic-viscous-plastic system (Hunke and Dukowicz (1997)). In this approach, $N_{EVP}$ small steps are carried out per ocean time step $\tau$. The procedure adopted in FESOM2 is a modified variant of the EVP approach (mEVP) written as

$$\alpha(\sigma_1^{p+1} - \sigma_1^p) = \frac{P_0}{\Delta^p + \Delta_{min}}(\dot{\epsilon}_1^p - \Delta^p) - \sigma_1^p, \tag{A7}$$

$$\alpha(\sigma_2^{p+1} - \sigma_2^p) = \frac{P_0}{(\Delta^p + \Delta_{min})e^2}\dot{\epsilon}_2^p - \sigma_2^p, \tag{A8}$$

$$\alpha(\sigma_{12}^{p+1} - \sigma_{12}^p) = \frac{P_0}{(\Delta^p + \Delta_{min})e^2}\dot{\epsilon}_{12}^p - \sigma_{12}^p, \tag{A9}$$

for stresses and

$$\beta(\boldsymbol{u}^{p+1} - \boldsymbol{u}^p) = -\boldsymbol{u}^{p+1} + \boldsymbol{u}^n - \Delta t \boldsymbol{f} \times \boldsymbol{u}^{p+1} + (\Delta t/m)[\boldsymbol{F}^{p+1} + a\boldsymbol{\tau} + C_d a\rho_o(\boldsymbol{u}_o^n - \boldsymbol{u}^{p+1})|\boldsymbol{u}_o^n - \boldsymbol{u}^p| - mg\nabla H^n] \tag{A10}$$

for momentum. Here, the shortcuts $\sigma_1 = \sigma_{11} + \sigma_{22}$ and $\sigma_2 = \sigma_{11} - \sigma_{22}$ are introduced for stresses and similarly for strain rates, $p = 1, \ldots, N_{EVP}$ is the index of subcycling, and $\alpha$ and $\beta$ are large parameters governing stability. Their values depend on mesh resolution and are determined experimentally. Different from the traditional EVP, mEVP is always stable provided $\alpha$ and $\beta$ are sufficiently large (e.g. $\alpha = \beta = 500$ were used in simulations reported in this paper). The number of substeps determines the closeness to a VP solution. Indeed, in the limit $p \to \infty$, when the difference between $p$ and $p+1$ estimates tends to zero, the equations above become the time discretized (A3) with (A4). The number $N_{EVP}$ needed for simulations is selected experimentally starting from $N_{EVP} \gg \alpha$ and reducing it to as low values as possible without a noticeable effect to solution. For meshes used in the evaluation here with the finest resolution about 4.5 km in the Arctic (fArc), $N_{EVP} = 100$ works well. We note that such small values of $N_{EVP}$ are only possible when using mEVP. Since halo exchange for the sea-ice velocity is needed on every iteration, low $N_{EVP}$ improves parallel scalability.

## Appendix B: Pipelined BiCGStab (PBiCGStab) algorithm for the SSH computation

**Algorithm 2** Preconditioned BiCGStab with maximum number of iterations $N_{\max}$ and residuum check for tolerance $r_{\text{tol}}$.

**Require:** $x_0$ initial guess; $b$ right hand side
**Require:** $A$ matrix; $M^{-1}$ preconditioner

  {Initialization}

  $r_0 = b - Ax_0; \quad p_0 = r_0$

  {Main loop}
  **for** $i = 0$ to $N_{\max}$ **do**
    **compute** $\hat{p}_i == M^{-1}p_i; \quad s_i = A\hat{p}_i$
    **begin reduce** $\langle r_0, r_i \rangle; \langle r_0, s_i \rangle$ **end reduce**
    $\alpha_i = \langle r_0, r_i \rangle / \langle r_0, s_i \rangle$
    $q_i = r_i - \alpha_i s_i$
    **compute** $\hat{q}_i == M^{-1}q_i; \quad y_i = A\hat{q}_i$
    **begin reduce** $\langle q_i, y_i \rangle; \langle y_i, y_i \rangle$ **end reduce**
    $\omega_i = \langle q_i, y_i \rangle / \langle y_i, y_i \rangle$
    $x_{i+1} = x_i + \alpha_i \hat{p}_i + \omega_i \hat{q}_i$
    $r_{i+1} = q_i - \omega_i y_i$
    **begin reduce** $\langle r_0, r_{i+1} \rangle; \langle r_{i+1}, r_{i+1} \rangle$ **end reduce**
    **convergence check: if** $\langle r_{i+1}, r_{i+1} \rangle \leq r_{\text{tol}}$ **then exit loop**
    $\beta_i = (\alpha_i \langle r_0, r_{i+1} \rangle) / (\omega_i \langle r_0, r_i \rangle)$
    $p_{i+1} = r_{i+1} + \beta_i(p_i - \omega_i s_i)$
  **end for**

---

**Algorithm 3** Preconditioned pipelined BiCGStab (Cools and Vanroose (2017)) as implemented in FESOM2 with maximum number of iterations $N_{\max}$ and residuum check for tolerance $r_{\text{tol}}$.

**Require:** $x_0$ initial guess; $b$ right hand side
**Require:** $A$ matrix; $M^{-1}$ preconditioner

  {Initialization}

  $r_0 = b - Ax_0; \quad \hat{r}_0 = M^{-1}r_0; \quad w_0 = A\hat{r}_0$
  **begin reduce** $\langle r_0, r_0 \rangle; \langle r_0, w_0 \rangle$
  **compute** $\hat{w}_0 = M^{-1}w_0; \quad t_0 = A\hat{w}_0$
  **end reduce**
  $\alpha_0 = \langle r_0, r_0 \rangle / \langle r_0, w_0 \rangle$
  $\beta_{-1} = 0, \quad \omega_{-1} = 0; \quad s_{-1} = 0; \quad \hat{s}_{-1} = 0$
  $z_{-1} = 0; \quad \hat{z}_{-1} = 0; \quad \hat{p}_{-1} = 0; \quad v_{-1} = 0$

  {Main loop}
  **for** $i = 0$ to $N_{\max}$ **do**
    $\hat{p}_i = \hat{r}_i + \beta_{i-1}(\hat{p}_{i-1} - \omega_{i-1}\hat{s}_{i-1}); \quad s_i = w_i + \beta_{i-1}(s_{i-1} - \omega_{i-1}z_{i-1})$
    $\hat{s}_i = \hat{w}_i + \beta_{i-1}(\hat{s}_{i-1} - \omega_{i-1}\hat{z}_{i-1}); \quad z_i = t_i + \beta_{i-1}(z_{i-1} - \omega_{i-1}v_{i-1})$
    $q_i = r_i - \alpha_i s_i; \quad \hat{q}_i = \hat{r}_i - \alpha_i \hat{s}_i; \quad y_i = w_i - \alpha_i z_i$
    **begin reduce** $\langle q_i, y_i \rangle; \langle y_i, y_i \rangle$
    **compute** $\hat{z}_i = M^{-1}z_i; \quad v_i = A\hat{z}_i$
    **end reduce**
    $\omega_i = \langle q_i, y_i \rangle / \langle y_i, y_i \rangle$
    $x_{i+1} = x_i + a_i \hat{p}_i + \omega_i \hat{q}_i$
    $r_{i+1} = q_i - \omega_i y_i; \quad \hat{r}_{i+1} = \hat{q}_i - \omega_i(\hat{w}_i - \alpha_i \hat{z}_i)$
    $w_{i+1} = y_i - \omega_i(t_i - \alpha_i v_i)$
    **begin reduce** $\langle r_o, r_{i+1} \rangle; \langle r_o, w_{i+1} \rangle; \langle r_o, s_i \rangle; \langle r_o, z_i \rangle; \langle r_{i+1}, r_{i+1} \rangle$
    **compute** $\hat{w}_{i+1} = M^{-1}w_{i+1}; \quad t_{i+1} = A\hat{w}_{i+1}$
    **end reduce**
    **convergence check: if** $\langle r_{i+1}, r_{i+1} \rangle \leq r_{\text{tol}}$ **then exit loop**
    $\beta_i = (\alpha_i \langle r_0, r_{i+1} \rangle) / (\omega_i \langle r_0, r_i \rangle)$
    $\alpha_{i+1} = \langle r_0, r_{i+1} \rangle / (\langle r_0, w_{i+1} \rangle + \beta_i \langle r_0, s_i \rangle - \beta_i \omega_i \langle r_0, z_i \rangle)$
  **end for**

## B1 Dependency of scaling results on time step, forcing, and ocean state.

Most of the scaling experiments in this paper were performed in accordance with the simple protocol, in which the model is initialised from a state of rest and run for a small number of time steps (1800 in our case). This design (hereafter called "cold start") allows us to easily conduct scaling experiments and permits quick testing of changes in the model code; no reading of the restart files is required that can become time consuming for large setups (making experiments very expensive in terms of CPU hours). Moreover, a similar strategy is used by many ocean and atmosphere modelling groups therefore greatly simplifying the comparison of the results. The downsides of this protocol are a smaller time step needed to keep high resolution setups stable and the ocean dynamics that is not fully developed yet. In this chapter, we explore consequences of these deficiencies for the results described in our manuscript.

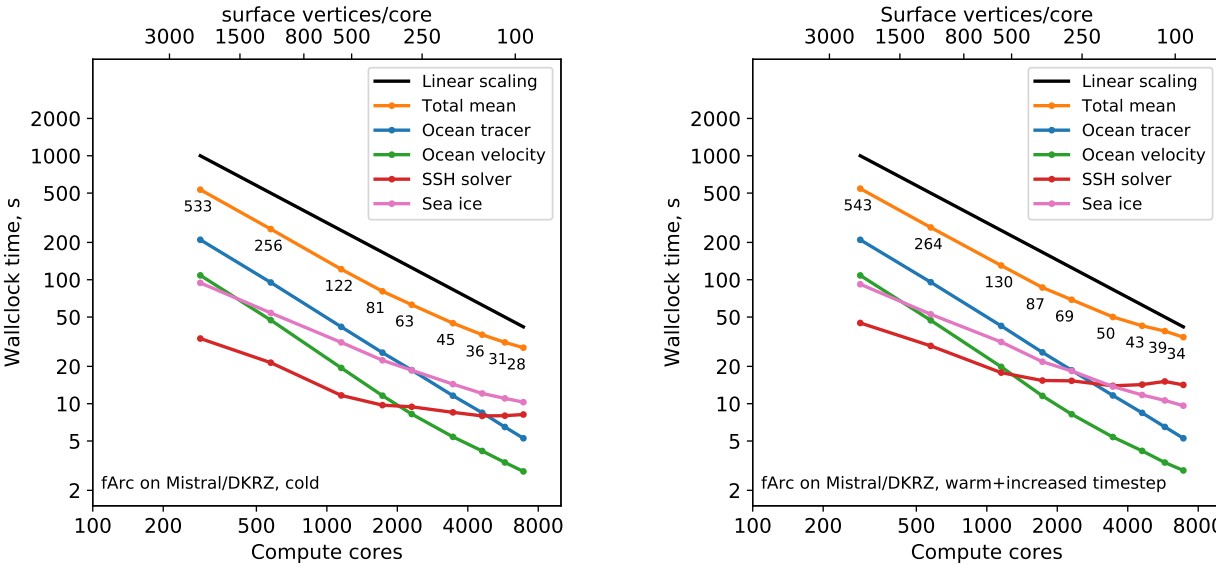

**Figure B1.** Scaling results for the fArc mesh on Mistral (DKRZ Hamburg) for "cold start" (left) and "warm start" with increased time step (right) experiments. The black line indicates linear scaling, the orange line (and the number labels) give the mean total computing time over the parallel partitions.

High resolution models have a several times larger time step in operational setting compared to the spin-up phase (that usually lasts for about a year). A larger time step makes the conditioning of the SSH matrix worse, hence there is an increase in the number of iterations in the SSH solver. To illustrate how this effect will influence the scaling behaviour we conducted two series of experiments for fArc mesh on Mistral/DKRZ – the "cold start" ones, with shorter time step (4 min) and "warm start" ones that start from the restart files (after one year spin-up) and use a larger time step (15 min), same as in fArc production

experiments (Fig. B1). The shape of the SSH solver curve stays practically the same but is shifted upwards. The increase in the total mean time is almost entirely due the computation time of the SSH solver (Fig. B2). Although the higher cost of the SSH solver matters, it only slightly changes the scaling for the total model mean. Note that deviations from linear scaling for total mean become visible only after 250 surface vertices per core – a number beyond our claimed break in linear scaling after around 300 vertices per core.

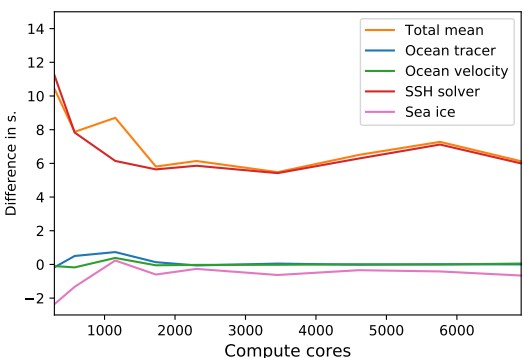

**Figure B2.** Difference between runtimes of the experiment with "warm start" and larger time step and the experiment with "cold start" and smaller time step.

To see how the number of barotropic solver iterations changes with time we perform experiments with the fArc mesh on Mistral/DKRZ using 5760 cores. The model is run from a state of rest (year 1948) and the Polar Science Center Hydrographic Climatology (PHC) (Steele et al., 2001) initial conditions for four consecutive years, and the number of solver iterations is recorded at every time step. Fig. B2 shows running mean values (window 500) of iteration counts. It is clear that changing ocean state and forcing has an effect on the number of SSH solver iterations and a full picture of the solver behaviour does not become apparent in short experiments as statistics over a longer period of time have to be collected. On the other hand, approximate values can be estimated from the short experiments, and the "cold start" experiments can probably give a lower bound estimate of the solver performance.

As a final note, one has to keep in mind that simple scaling experiments described in the manuscript can explore mostly the scalability of model components, while clearly absolute numbers can vary in a wide range. For example, the sea-ice model will show different runtimes depending on the number of EVP subcycles (see e.g. Koldunov et al. (2019)). The runtimes for tracers can be reduced nearly two-fold if the flux corrected transport (FCT) option is off (it is on now), or can be increased if higher-order schemes are used. Scaling will be improved if the number of layers is increased, and so on. The scaling of the total mean will be affected by these details, thus any discussion can only be conducted in a certain approximate sense.

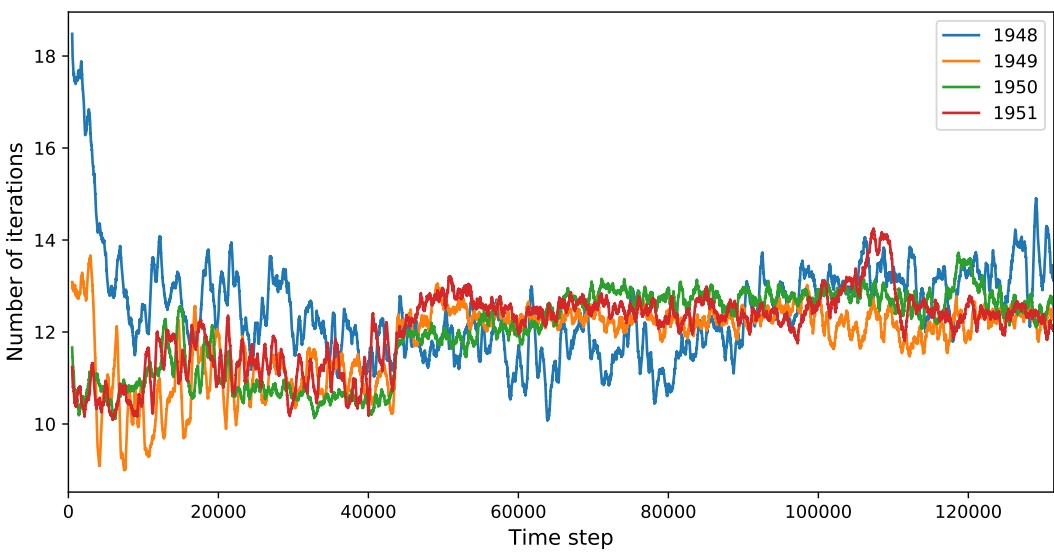

**Figure B3.** Number of barotropic solver iterations at each time step for four consecutive years. The year 1948 begin with the "cold start". Running mean smoothing with window 1000 is applied. Ocean model time step - 4 min.

*Author contributions.* SD and VA conceived the original idea for this maniscript. PS, DS, and NR developed the operational version of the model. NR implemented the new version of the linear solver for the SSH. VA implemented hierarchical partitioning. NK carried out experiments on Mistral and JUWELS, NR carried out experiments on Ollie, VA carried out experiments on Emmy. NK and NR produced graphs. SD, NK, VA, and NR wrote the initial draft of the manuscript. All co-authors contributed to the final draft of the manuscript. TJ supervised the project.

*Competing interests.* The authors declare that they have no conflict of interest.

*Acknowledgements.* We thank Peter Düben, Mark Petersen, and Leonidas Linardakis for their very helpful comments. We also thank Marshall L. Ward for providing throughtput numbers for different models. This paper is a contribution to the projects S1 (Diagnosis and Metrics in Climate Models) and S2 (Improved parameterisations and numerics in climate models) of the Collaborative Research Centre TRR 181 "Energy Transfer in Atmosphere and Ocean" funded by the Deutsche Forschungsgemeinschaft (DFG, German Research Foundation) - Projektnummer 274762653. D. Sidorenko is funded by the Helmholtz Climate Initiative REKLIM (Regional Climate Change). The runs were performed at Deutsche Klimarechenzentrum (DKRZ), Jülich Supercomputing Centre (JSC), Regionales Rechenzentrum Erlangen (RRZE), and AWI Computing Centre. We would also like to extend special thanks to Dr. Hendryk Bockelmann (DKRZ) for his active support with scaling experiments on large numbers of nodes.

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
