# Peer review of "Scalability and some optimization of the Finite-volumE Sea-ice-Ocean Model, Version 2.0 (FESOM2)"

_Geoscientific Model Development, 2018_

## Referee Comment (RC1) · Peter Düben (Referee) · 25 Apr 2019

This is an excellent paper that should be published. The results are important for the community as current ocean models are struggling to make efficient use of modern supercomputers. It is an excellent result that a model based on unstructured grids can compete with structured models in terms of throughput on large supercomputers. The challenges for scalability that have been identified and quantified in this papers (2D models) should be taken very seriously by the community.

The paper could essentially be published as it is but may improve if the minor comments below are addressed. Non of this is mandatory.

[Figure]

- There are minor (but many) problems with the English language throughout the paper that should be corrected. In particular regarding the use of "the".

- p.3 l. 26: "will be lower" Why is this?

- Section 2.2: Can you briefly state the physical motivation why these grids were generated with the focus on resolution in very different locations?

- Section 3: This may not be possible but it would be great if you could discuss/speculate a bit more how the optimal supercomputer for FESOM would look like if assembled from existing hardware components.

- p.9 l.23: I do not understand why.

- p.13 l.8: "depends on the sizes of parallel partitions" Really? Why?

- p.16 l.5+6: Why are these "estimates"? I guess these have been measured?

- Figure 11 (left): I am not sure whether this level of detail is required here. You could just add 1-2 sentences of text and remove this figure.

- Section 6.3: You could cite the ESCAPE project that is trying to develop dwarfs for weather and climate models. You could just refer to the webpage http://www.hpc-escape2.eu/ .?

- You could maybe improve the discussion on the use of concurrency to achieve further improvements in scalability. Expected benefit, how to apply it, where to start,...

---

## Referee Comment (RC2) · Leonidas Linardakis (Referee) · 28 Apr 2019

This paper falls under the category of "model description". The performance on 2D unstructured grids has been known to be on par with the structured ones in the small community of unstructured grid modeling, but has not been publiced in a comprehensive way for GC models. This paper serves the purpose of discussing and communicating this knowledge, demonstrating that an operational unstructured-grid ocean model is on par, performance-wise, with structured-grid ones. I have major concerns though about the protocol followed in the design of the scaling experiments, and the numbers obtained from them, so I recommend the paper to be published after a major revision.

1. The authors present two sets of scaling experiments: an "idealized" set and a "realistic" one, and they discuss their differences at page 17, lines 7-14. In the idealized setup the SSH solver is four times faster than the realistic one (STORM experiment on 13824 cores), which renders the related scaling numbers and analysis of the idealized setup at the very least questionable. On the other hand, the realistic setup is presented under the title "Testing throughput in an operational configuration", implying that this is an operational setup. The original operational STORM setup though is very different than the one described by the authors, with a large number of specific diagnostics and high frequency output, rendering the term "operational" questionable. I propose a single set of realistic experiments, that demonstrate the model's scaling characteristics in realistic conditions, but no "operational" ones. As the authors point out, the cost of I/O and diagnostics in operational setups varies widely with the requirements, and does not reveal the performance of the core model, which is the interest in the scaling analysis of the model. In the model scaling experiments I/O and additional diagnostics are not needed. The experiments should be restarted from a one year run; running for the duration of 15days – 1month should be sufficient.

2. The information for the model and experiment setups is scattered, and I found it difficult to piece it together. A table would be useful, that describes the parametrizations used, type of of vertical coordinates, solver tolerance, timestep, run duration, etc.

3. A very interesting and important point is raised on page 12, lines 25 – 30: how to tune the solver so it provides the required quality at a minimal cost. It would be interesting to know how the authors define "robust results" (line 25), what is the analysis and the metrics (max error, biases, conservation) that they used for this tuning.

4. A minor comment: In Fig 10, why the number of the solver iterations depend on the number of cores?

5. I recommend to cite: A. E. MacDonald et al. "A general method for modeling on irregular grids", 2010, doi:10.1177/1094342010385019

---

## Short Comment (SC1) · 29 Apr 2019

Hi, I am the author of the Ward (2016) presentation cited in the Table 3. Overall the results in this paper are very nice, and am looking forward to learning more about the model. My comments are only restricted to the timings used in Table 3.

My main comment is that the numbers which I presented at ECMWF were attempting to assess the scalability of existing configurations of NEMO, MOM5, and MOM6 in use on our system(Raijin at NCI Australia), and were not necessarily representative measures of performance of the models.

[Figure]

For example, the NEMO configuration was quite old even at the time (v3.4), and two attendees pointed out potential implementation issues of the tripole that were resolved in v3.6.

Additionally, I have my own internal numbers showing that MOM5 0.25 can scale much higher than 25 yr/day. For example, I have shown results showing performance of sea-ice (SIS) bound timings of 35 yr/day, and can increase this to nearly 60 yr/day, when sea ice is running concurrently to MOM5 and we allow high inefficiency.

(This is my usual "reference", a presentation at AMOS in 2016 where the ocean model was individually profiled:

https://marshallward.org/talks/amos2016.html#/11/

though again it's just a presentation and hasn't necessarily been stress tested.)

I know how difficult it can be to find cited performance numbers for ocean models, and I have probably not published enough of my own numbers, but feel a bit uncomfortable being associated with these particular numbers and wonder if they could be replaced or revised in some way.

I do have one published paper of MOM performance numbers which may be suitable:

* https://link.springer.com/chapter/10.1007/978-3-319-15994-2_55

And Balaji et al's CPMIP paper may provide some useful comparison numbers:

* https://doi.org/10.5194/gmd-10-19-2017

Or, if you do not want to change the reference, maybe just disclose somewhere that the numbers are not necessarily reflective of peak performance but "typical" performance, in which case I'd request that you use the lower ~1000 CPU timings.

---

## Referee Comment (RC3) · Mark Petersen (Referee) · 6 May 2019

This paper evaluates the performance characteristics of FESOM 2.0 using three resolutions and four machines. It will be a great reference for all ocean model developers, because it explains the root causes of ocean /sea ice bottlenecks, and discusses the options available to those designing these algorithms. The two slow-downs are the SSH solver and the sea ice because they are two dimensional yet require frequent communication, so require a large number of small MPI messages. My group struggles with exactly the same issues, and we have gone back and forth between implicit and split-explicit barotropic solvers over the years in POP and MPAS-Ocean. In fact,

we are currently testing improved filtering methods for split-explicit as well as implicit methods in MPAS-Ocean. Figures 2-4 beautifully illustrate the crux of the matter: these two stubborn pieces don't scale.

The paper is very well written. The introduction provides useful context and motivation for the topic. Later sections clearly lay out the choices available to the developer and the corresponding results. The algorithm write-ups are a concise summary of complex code. The authors paid attention to important details – for example, the scaling plots are very well labeled and are easy to interpret, even while displaying a lot of information. I particularly appreciate table 3, as it gives us a rough idea of performance comparisons between models, despite the potential mis-matches between models. The performance of FESOM 2.0 is very impressive, and sets a high bar for throughput for other ocean models.

Table 3. It is possible to go a step further, and compute SYPD/vertices/levels/cores or SYPD/(active 3D cells)/cores. We know that the 2D solvers are a good fraction, so SYPD/vertices/cores is also helpful. I find myself trying to estimate comparisons by eye, taking into account cores and layers. The addition of these columns, though still an imperfect comparison, would be helpful. For a single model with perfect strong scaling and perfect weak scaling, this is a single number. Indeed, when I've made plots of these values in the past for multiple resolutions and core counts, everything collapses to a single number within a reasonable range of core counts. The exception is the factor of the number of layers, which doesn't scale linearly with SYPD. The number of degrees of freedom (DOF) is also different between triangle, quad, and mostly-hex meshes, so I often normalize by the number of prognostic DOF the model solves for, i.e. (number tracers)*(number tracer points) + (number of velocity DOF) For example, a quad B-grid like POP has two velocity DOF per cell, but a hex C-grid has three.

A design choice of whether to solve an extra equation for thickness also influences performance. In a pure z-level model like POP, only the top layer thickness varies for the free surface, so almost no computation is required – we only compute vertical transport

by solving the continuity equation. Once we moved to a thickness equation and ALE in MPAS-Ocean, we actually solve for the thickness as an extra tracer (effectively one) using identical algorithms, in order to ensure consistency between advection of tracers and thickness. This is mathematically elegant and truer to the conservation equations, but I suspect it is a waste of computation because the ALE routine has already decided where the layer interfaces should be placed (z-star is similar, since thickness may simply be diagnosed). Of course, these inefficiencies may be a worthwhile price to pay to make the code more general, because it is now easier to implement a non-Boussinesq code where we advect mass rather than thickness (i.e. area-normalized volume). It is also now feasible to use true ALE and only remap every n time steps. For the purpose of this performance paper, it would be useful to know if you had similar considerations, and what factors influenced your decisions. Do you solve a thickness equation (included in tracer timers) or do you simply diagnose layer thickness?

In figures 2-4 the total is split into four pieces, which is perfect for those purposes. It would be useful to have another plot that splits ocean tracer and ocean velocity into their smaller components. For example, within ocean tracer, what is the percentage of each: advection, determining vertical mixing coefficients, implicit vertical mixing, GM, equation of state, etc. As a fellow ocean model developer, this helps me find potential improvements in my code, and I could publish a similar plot for comparison. A bar chart would illustrate this quite well. This could all be done as a sample from just one domain, one machine, and one node count, as I think it does not vary much for these two that scale so well.

For the SSH solver and sea ice, we have separate sub-timers around communication and computation. An additional scaling plot with these two separated out would show the communication problem even more strongly, as it should all be in the communication. Again, a single machine and domain would illustrate the point.

Why did you choose 47 layers? Layers that are a multiple of 8 or 16 are often recommended for memory access and vectorization (and you are very close!). In fact,

there is a method called strip-mining where you always loop down to a multiple of 8 in the vertical, and multiply by a mask for land cells below ocean, just to get the cache alignment. Did you consider any of these issues in your original FESOM design or performance improvements?

On a related note, it's worth mentioning in this paper the index order and your reasoning for it, as it is an important performance consideration. I'm sure you mentioned it in some other papers, but it is relevant here too. I assume all unstructured models use the vertical as a tighter index than the horizontal, as that's "all we've got" for exact memory alignment. A secondary choice is whether to make a 'super array' for tracers, where you use a tracer index to loop through all the tracer variables, rather than individual tracer variables. If you did this, what order did you place the indices, and why?

Did you consider trying to make your halos wider for less frequent communication? That doesn't work for your implicit SSH solve if it requires a global communication, but it is a possibility in other parts of the code. Would be good to comment on this design choice in the text.

My other comments are very minor. I've attached a copy with grammatical corrections. 1. Does fArc mean something? 2. Fig 7, 9: Add perfect scaling line 3. Table 3. Label unit for time step

Please also note the supplement to this comment:

[revised manuscript text omitted]
 (ua) = S_a, \quad \partial_t h_{ice} + \nabla \cdot (uh_{ice}) = S_{ice}, \quad \partial_t h_s + \nabla \cdot (uh_s) = S_s \tag{A2}$$

with $S_a$, $S_{ice}$ the sources related to sea sea-ice melting and freezing, and $S_s$ the sources due to snow precipitation and melting; as well as the momentum equation

$$m\,\partial_t u_{ice} = a\tau - mfk \times u_{ice} - aC_d\rho_w(u_{ice} - u)|u_{ice} - u| + \nabla \cdot \sigma - mg\nabla\eta, \tag{A3}$$

where $u_{ice} = (u_{ice}, v_{ice})$ is the sea-ice velocity, $m = \rho_{ice}h_{ice} + \rho_s h_s$ the combined mass of sea-ice and snow per unit area, $\rho_{ice}$ and $\rho_s$ the densities of sea-ice and snow, respectively, $f$ the Coriolis coefficient, $k$ the unit vertical vector, $C_d$ the ice-ocean drag coefficient, $\rho_w$ the water density, $u$ the ocean velocity at the ocean-ice interface, $\tau$ the wind stress at the sea-ice surface, and $\sigma = \{\sigma_{ij}\}$ the internal ice stress tensor computed using the viscous-plastic (VP) rheology (Hibler (1979))

$$\sigma_{ij} = 2\eta(\dot{\epsilon}_{ij} - (1/2)\delta_{ij}\dot{\epsilon}_{kk}) + \zeta\delta_{ij}\dot{\epsilon}_{kk} - (1/2)\delta_{ij}P, \tag{A4}$$

where $\delta_{ij}$ is the usual Kronecker delta, $\eta$, $\zeta$ are the moduli ('viscosities'), and $P$ is the sea-ice strength. The strain rate tensor $\dot{\epsilon}$ is defined as

$$\dot{\epsilon}_{11} = \partial u/\partial x, \quad \dot{\epsilon}_{22} = \partial v/\partial y, \quad \dot{\epsilon}_{12} = \dot{\epsilon}_{21}\dot{\epsilon}_{21} = (1/2)(\partial u/\partial y + \partial v/\partial x).$$

The sea-ice strength $P$ and viscosities are computed as

$$P = P_0, \quad \zeta = (P_0/2)/(\Delta + \Delta_{min}), \quad \eta = \zeta/e^2, \tag{A5}$$

where

$$P_0 = h_{ice}p^*\exp{-C(1-a)}, \quad \Delta^2 = (\dot{\epsilon}_{11}^2 + \dot{\epsilon}_{22}^2)(1 + 1/e^2) + 4\dot{\epsilon}_{12}^2/e^2 + 2\dot{\epsilon}_{11}\dot{\epsilon}_{22}(1 - 1/e^2), \tag{A6}$$

$e = 2$ and $C = 20$; as the default values in FESOM, we set $\Delta_{min} = 2 \cdot 10^{-9}$ s$^{-1}$ and $p^* = 27500$ N/m$^2$.

The main difficulty of solving (A3) with (A4) is that typical values of viscosities are very large leading to a very stiff and highly nonlinear system. In most cases it is solved by augmenting (A4) with pseudo-elasticity, which results in a elastic-viscous-plastic system (Hunke and Dukowicz (1997)). In this approach, $N_{EVP}$ small steps are carried out per ocean time step $\tau$. The procedure adopted in FESOM is a modified variant of the EVP approach (mEVP) written as

$$\alpha(\sigma_1^{p+1} - \sigma_1^p) = \frac{P_0}{\Delta^p + \Delta_{min}}(\dot{\epsilon}_1^p - \Delta^p) - \sigma_1^p, \tag{A7}$$

$$\alpha(\sigma_2^{p+1} - \sigma_2^p) = \frac{P_0}{(\Delta^p + \Delta_{min})e^2}\dot{\epsilon}_2^p - \sigma_2^p, \tag{A8}$$

$$\alpha(\sigma_{12}^{p+1} - \sigma_{12}^p) = \frac{P_0}{(\Delta^p + \Delta_{min})e^2}\dot{\epsilon}_{12}^p - \sigma_{12}^p, \tag{A9}$$

for stresses and

$$\beta(u^{p+1} - u^p) = -u^{p+1} + u^n - \Delta t f \times u^{p+1} + (\Delta t/m)[F^{p+1} + a\tau + C_d a\rho_o(u_o^n - u^{p+1})|u_o^n - 
[revised manuscript text omitted]

---

## Author Comment (AC1) · 19 Jul 2019

**Referee 1 (Peter Dueben)**

**• There are minor (but many) problems with the English language throughout the paper that should be corrected. In particular regarding the use of "the".**
*We went over the paper and did our best to correct the language. We also greatly benefited from corrections suggested by you and other reviewers.*

**• p.3 l. 26: "will be lower" Why is this?**
*The computational part in unstructured models tends to be slower than in the structured models, the communication parts are approximately equal, then the ratio of communication to computation must be lower as well.*

**• Section 2.2: Can you briefly state the physical motivation why these grids were generated with the focus on resolution in very different locations?**
*We have added the following passages to the text:*
*"...a low resolution CORE2 mesh specially constructed to better represent global circulation in a low resolution setup."*
*"...mesh, referred to as fArc (FESOM Arctic), aims to better represent circulation in the Arctic Ocean while maintaining a relatively low resolution in the rest of the ocean."*

**• Section 3: This may not be possible but it would be great if you could discuss/speculate a bit more how the optimal supercomputer for FESOM would look like if assembled from existing hardware components.**
*This might be very wide off the target, but the current CPU architectures appear to be pretty good for nearly all 3D computational parts of FESOM2. The most pressing bottlenecks are the I/O and the 2D parts such as the sea ice and the SSH. Thus the potential for improvement seems to lie in the direction of improved memory bandwidth, lower communication latency and more efficient file systems -- this can be then used as an indication when choosing 'optimal' hardware.*
*We have added slightly modified version of this answer to the conclusions:*
*"The current CPU architectures appear to be well-suited for nearly all 3D computational parts of FESOM2, thus the potential for improvement seems to lie in the direction of improved memory bandwidth, lower communication latency and more efficient file systems – this can be then used as an indication*
*when choosing 'optimal' hardware."*

**• p.9 l.23: I do not understand why.**
*The sentence now reads as follows:*
*"This good scaling of the 3D parts of the FESOM2 code suggests that the 3D model parts may be efficiently computed on hardware architectures with low memory per computational core (e.g. GPUs)."*

**• p.13 l.8: "depends on the sizes of parallel partitions" Really? Why?**
*The limiting factor is the convergence of the RAS preconditioner whose efficiency depends on the sizes of parallel partitions" (See Smith, Bjørstad, Gropp, 2004). We added this reference to the text.*

**• p.16 l.5+6: Why are these "estimates"? I guess these have been measured?**

*These are estimates since we did not run our scaling tests for the whole year (as opposed to our 'operational' runs). Instead, we just estimated the throughput based on timings for 1800 time steps and time step sizes.*

**• Figure 11 (left): I am not sure whether this level of detail is required here. You could just add 1-2 sentences of text and remove this figure.**

*We believe this figure can be illustrative for the non-HPC readers, so we intend to keep it.*

**• Section 6.3: You could cite the ESCAPE project that is trying to develop dwarfs for weather and climate models. You could just refer to the webpage http://www.hpcescape2.eu/ .?**

*We have added the following sentence: "In the framework of the currently funded EU project ESCAPE, dwarfs for several weather and climate models are being actively developed (http://www.hpc-escape2.eu/, last access: 19 June 2019)."*

**• You could maybe improve the discussion on the use of concurrency to achieve further improvements in scalability. Expected benefit, how to apply it, where to start,...**

*A) Concurrency for halo exchange*
*A simple version with the pseudocode:*
  *compute variable A on nodes*
  *exchange_nod_begin(A)      -> wrapper to MPI_Isend, MPI_Irecv*
  *compute B, independent of outer halo of A*
  *exchange_nod_end(A)        -> wrapper to MPI_Wait*
*is already used in FESOM2 wherever it is simple to implement.*
*We plan to introduce re-sorting of the local fields into inner nodes and inner halo nodes (being sent) in the same array, to be able to split loops like this:*
  *compute variable A on inner halo*
  *exchange_nod_begin(A)*
  *compute_variable A on inner non-halo nodes*
  *exchange_nod_end(A)*
  *compute B dependent on outer halo of A*
*The ice solver should benefit the most, because of a poor computation to communication ratio, and only little overlap of the simple kind. pARMS already re-sorts the local fields and overlaps halo exchange with computation during the matrix-vector product.*

*B) Concurrency for global reduction*
*We implemented this into the SSH solver as explained in the paper.*
*We decided not to include this discussion in the manuscript --- it is a bit too technical and we do not have at present any quantitative estimates of benefits. We, however, mention briefly these issues in the section 'SSH strategy'.*
* * *
**Referee 2 (Leonidas Linardakis)**

**1. The authors present two sets of scaling experiments: an "idealized" set and a "realistic" one, and they discuss their differences at page 17, lines 7-14. In the idealized setup the SSH solver is four times faster than the realistic one (STORM experiment on 13824 cores), which renders the related scaling numbers and analysis of the idealized setup at the very least questionable. On the other hand, the realistic setup is presented under the title "Testing throughput in an operational configuration", implying that this is an operational setup. The original operational STORM setup though is very different than the one described by the authors, with a large number of specific diagnostics and high frequency output, rendering the term "operational" questionable. I propose a single set of realistic experiments, that demonstrate the model's scaling characteristics in realistic conditions, but no "operational" ones. As the authors point out, the cost of I/O and diagnostics in operational setups varies widely with the requirements, and does not reveal the performance of the core model, which is the interest in the scaling analysis of the model. In the model scaling experiments I/O and additional diagnostics are not needed. The experiments should be restarted from a one year run; running for the duration of 15days – 1month should be sufficient.**

*The main difference between our "operational" and idealized experiments (excluding IO) is the time step which is larger in the "operational" case (a smaller time step is used in the initial spin up). This makes conditioning of the SSH matrix worse, hence the increase in the number of iterations in the SSH solver. We overlooked this aspect in the original version of the manuscript, attributing the increase to dynamics. While dynamics is important, it is not the main reason. We did not have an opportunity to rerun experiments with the "operational" time step, but we did this for one test case to show that the cost of all other components remains the same, and it is only the SSH solver that becomes more expensive. We show now that although the higher cost of the SSH solver matters, it only slightly changes the scaling for the total model mean. The reason is that the parallel partition size for which the cost of the SSH solver becomes greater than that of sea-ice solver or tracers only occurs at 250 vertices per core or less. We added a corresponding explanation to the manuscript (Appendix B) and kept the previous results. We hope that our explanations are sufficient to interpret our results correctly. Below we provide more clarifications.*

*We did two series of experiments for fArc at Mistral/DKRZ - the "idealized" ones, with cold start and shorter time step (4 min) and "operational" ones that start from the restart files and longer time step (15 min) that is used in fArc production experiments (Fig. R1). The shape of the SSH solver curve stays practically the same but is shifted upwards. The increase in total mean time is almost entirely due to increase in the computation time of the SSH solver (Fig. R2). Note that deviations from linear scaling for total mean become visible after 250 surface vertices per core, which is beyond our claim.*

[Figure]

*Fig. R1 Scaling results for the fArc mesh on Mistral (DKRZ Hamburg) for "Cold start" (left) and "Warm Start" with increased time step (right) experiments. The black line indicates linear scaling, the orange line (and the number labels) give the mean total computing time over the parallel partitions.*

[Figure]

*Fig. R2 Difference between run times of experiment with warm start and large time step and experiment with cold start and small time step.*

*Also note that we explore mostly the scalability of model components (i.e. relative changes), while clearly absolute numbers can vary in a wide range. For example, the sea ice model will show different runtimes depending on the number of EVP subcycles (see e.g. Koldunov et al., 2019). The runtimes for tracers can be reduced nearly two-fold if the flux corrected transport (FCT) option is off (it is on now), or can be increased if higher-order schemes are used. Scaling will be improved if the number of layers is increased, and so on. The scaling of total mean will be affected by these details and can only be discussed in certain approximate sense.*

**2. The information for the model and experiment setups is scattered, and I found it difficult to piece it together. A table would be useful, that describes the parametrizations used, type of vertical coordinates, solver tolerance, timestep, run duration, etc.**

*We have decided against an extra table since most of the parameters (except for time step sizes) are the same. Instead, we expand the "Test cases and HPC systems used in the study" Section with the following text:*

*"All meshes have 47 unevenly spaced vertical layers. The K-profile parameterization (Large et al., 1994) is used for the vertical mixing, and isoneutral diffusion (Redi, 1982) and the Gent–McWilliams (GM) parameterization (Gent and McWilliams, 1990) are utilized. Note that the 5 GM coefficient is set to 0 when the horizontal grid spacing goes below 25 km. The horizontal advection scheme for tracers uses a combination of third and fourth-order fluxes with flux corrected transport (FCT), for horizontal*
*momentum advection, a second-order flux form is used. The Leith viscosity (Leith, 1968, 1996) is used together with the modified Leith viscosity in combination with weak biharmonic viscosity. The vertical advection scheme for tracers and momentum combines third and fourth order fluxes. Sea-ice dynamics uses the mEVP option (Kimmritz et al., 2017; Koldunov*
*et al., 2019) with 100 subcycles. To ensure model stability for "cold start" experiments with high resolution meshes (fArc and STORM), we use smaller time steps (4 and 2 minutes respectively) compared to the time steps used in production runs (15 and 10 minutes respectively). The time step of 45 minutes is used in all COREII experiments."*

**3. A very interesting and important point is raised on page 12, lines 25 – 30: how to tune the solver so it provides the required quality at a minimal cost. It would be interesting to know how the authors define "robust results" (line 25), what is the analysis and the metrics (max error, biases, conservation) that they used for this tuning.**

*In the early days of FESOM, the first tuning was done using PETSc leading to our choice of BiCGstab, restricted additive Schwarz, and local ILU. To improve portability, we switched later to pARMS adding missing features (BiCGstab) and some more advanced schemes such as Schur complement combined with RAS, but nevertheless, BiCGstab + simple RAS + ILU gave the best performance. As criteria we tested different FESOM1 (and FESOM2) setups and observed convergence rates. In addition, this combination was used in a variety of FESOM1 and FESOM2 settings observing the robustness.*

*Our interpretation of robustness means that the number of iterations needed to reach tolerance does not show large variations. The acceptable error tolerance is an experimental parameter. It can be estimated by running simplified setups such as, for example, the linear Munk gyre, where an analytical solution is available. Many (unpublished) tests were done in earlier days of FESOM to see that very little improvement is possible beyond the tolerances of $10^{-8}$ currently used by us.*

*Approximate solutions are always conservative, which can be traced back to how the solution is constructed in the Krylov subspace methods. For example, in the linear free surface option used by us in the manuscript, the area weighted sum of elevation stays zero*

*because the rhs has the same property. The rhs satisfies this property because it has the form of the divergence of fluxes.*

*We feel that this information exceed the scope of the manuscript and decided not to include it. However, we would be willing to share our experience if requested.*

**4. A minor comment: In Fig 10, why the number of the solver iterations depend on the number of cores?**
*The factor is the convergence of the RAS preconditioner whose efficiency depends on the sizes of parallel partitions (See Smith, Bjørstad, Gropp, 1996).*

**5. I recommend to cite: A. E. MacDonald et al. "A general method for modeling on irregular grids", 2010, doi:10.1177/1094342010385019**

*We have added the following sentence to "SSH strategy" section: "The optimal data structure for unstructured meshes is still under discussion \citep{MacDonald2011}"*
* * *
**Referee 3 (Mark Petersen)**

**• Table 3. It is possible to go a step further, and compute SYPD/vertices/levels/cores or SYPD/(active 3D cells)/cores. We know that the 2D solvers are a good fraction, so SYPD/vertices/cores is also helpful. I find myself trying to estimate comparisons by eye, taking into account cores and layers. The addition of these columns, though still an imperfect comparison, would be helpful. For a single model with perfect strong scaling and perfect weak scaling, this is a single number. Indeed, when I've made plots of these values in the past for multiple resolutions and core counts, everything collapses to a single number within a reasonable range of core counts. The exception is the factor of the number of layers, which doesn't scale linearly with SYPD. The number of degrees of freedom (DOF) is also different between triangle, quad, and mostly-hex meshes, so I often normalize by the number of prognostic DOF the model solves for, i.e. (number tracers)*(number tracer points) + (number of velocity DOF) For example, a quad B-grid like POP has two velocity DOF per cell, but a hex C-grid has three.**
*Thank you for this suggestion! We have follower your approach and and added the following text:*
*"The relationship between resulting SYPD, basic model characteristics, and computer resources can be expressed as:*

$$SYPD =c_{SYPD}\frac{\Delta t * N_{core}}{N_{3D}}, $$
*so the SYPD is directly proportional to model time step ($\Delta t$) and number of computational cores ($N_{core}$), while inversely proportional to the number of degrees of freedom ($N_{3D}$). The larger the proportionality constant $c_{SYPD}$ the more computationally efficient is the model. In Table~\ref{tab:throughtput}, we list $c_{SYPD}$ for*

*3D (vertices\*number of layers) and 2D degrees of freedom (vertices) since 2D model parts solvers take up a~significant fraction of computational time."*

*We have provided numbers of $c_{SYPD}3D$ and $c_{SYPD}2D$ in the table.*

**• A design choice of whether to solve an extra equation for thickness also influences performance. In a pure z-level model like POP, only the top layer thickness varies for the free surface, so almost no computation is required – we only compute vertical transport by solving the continuity equation. Once we moved to a thickness equation and ALE in MPAS-Ocean, we actually solve for the thickness as an extra tracer (effectively one) using identical algorithms, in order to ensure consistency between advection of tracers and thickness. This is mathematically elegant and truer to the conservation equations, but I suspect it is a waste of computation because the ALE routine has already decided where the layer interfaces should be placed (z-star is similar, since thickness may simply be diagnosed). Of course, these inefficiencies may be a worthwhile price to pay to make the code more general, because it is now easier to implement a non-Boussinesq code where we advect mass rather than thickness (i.e. area-normalized volume). It is also now feasible to use true ALE and only remap every n time steps. For the purpose of this performance paper, it would be useful to know if you had similar considerations, and what factors influenced your decisions. Do you solve a thickness equation (included in tracer timers) or do you simply diagnose layer thickness?**
*Setups used in the experiments described in the manuscript use linear free surface, so the overhead of ALE is virtually absent. In our experience, using ALE in z\* option increases the run time within 10%. Although advection of thickness takes some time, it is relatively cheap because no fancy reconstruction is used in this case as opposed to tracers. The replacement of one tracer advection scheme with another one or switching on/off the GM parameterization generally affects the runtime even more than the ALE, so it is within the "error bars". Since updating layer thicknesses is a three-dimensional procedure, it is not expected to destroy scalability. So yes, in our opinion, the benefits of having the ALE are worth its cost. In the future, we are willing to explore the advantages/disadvantages of implementing the ALE via remapping, for it promises to remove CFL limitations of vertical advection which are very annoying.*

**• In figures 2-4 the total is split into four pieces, which is perfect for those purposes. It would be useful to have another plot that splits ocean tracer and ocean velocity into their smaller components. For example, within ocean tracer, what is the percentage of each: advection, determining vertical mixing coefficients, implicit vertical mixing, GM, equation of state, etc. As a fellow ocean model developer, this helps me find potential improvements in my code, and I could publish a similar plot for comparison. A bar chart would illustrate this quite well. This could all be done as a sample from just one domain, one machine, and one node count, as I think it does not vary much for these two that scale so well.**
*We have added the plot with more detailed split in components.*

[Figure]

Fig. R3 Mean wallclock runtimes for different model components. Experiment run for one year (11680 time steps on CORE2 mesh, 288 cores on Ollie/AWI system with time step of 45 minutes. Pressure includes the estimate of the equation of state, and slope includes computations of isoneutral slope.

The following text is added:
"Runtime values detailing performance of individual model components are presented in Fig. 5. Although obtained in simulations using the CORE2 mesh on 288 cores of Ollie/AWI and simulating 11680 time steps (one model year with 45 minute time step), their relative values are representative for other meshes."

**• For the SSH solver and sea ice, we have separate sub-timers around communication and computation. An additional scaling plot with these two separated out would show the communication problem even more strongly, as it should all be in the communication. Again, a single machine and domain would illustrate the point.**

*We provided graphs showing profiling of the sea-ice routines for 36 and 72 cores, which indicate that already for 72 cores (and relatively large partitions) time spent in communications takes a noticeable part of the ice step. We did not measure the time spent in communication in our simulations because our communications are hidden inside ice routines, and we were measuring times on the level of subroutines. Since, in our case, the sea-ice and ocean are run on the same partitioning, there are many idle cores that do not do computations, and direct measurement of time spent on non-blocking communications might be misleading. We work on respective diagnostics and will inform the reviewer separately.*

**• Why did you choose 47 layers? Layers that are a multiple of 8 or 16 are often recommended for memory access and vectorization (and you are very close!). In fact, there is a method called strip-mining where you always loop down to a multiple of 8 in the vertical, and multiply by a mask for land cells below ocean, just to get the cache alignment. Did you consider any of these issues in your original FESOM design or performance improvements?**

*The number is taken for historical reasons, for it was used in the earlier finite-element version of FESOM. In the present version of FESOM2.0, our vertical loops always go to the local bottom, so we deal with the variable number of layers and not necessarily benefit from making the total number of levels multiple of 8. We are considering the option mentioned, but it is a subject of future work.*

**• On a related note, it's worth mentioning in this paper the index order and your reasoning for it, as it is an important performance consideration. I'm sure you mentioned it in some other papers, but it is relevant here too. I assume all unstructured models use the vertical as a tighter index than the horizontal, as that's "all we've got" for exact memory alignment.**

*FESOM2.0 uses vertical as a tighter index. There are two aspects. First, this allows re-using the neighborhood information along the vertical column, which is not the case if the horizontal index is a tighter one. This aspect is important: two-dimensional shallow-water unstructured-mesh codes are much slower than their structured-mesh counterparts for they need to assess auxiliary look-up tables all the time. In unstructured-mesh 3D codes, the more vertical layers we use the closer we are in performance to structured-mesh models. Second, it is memory alignment for vertical operations and sufficient cache efficiency for horizontal operations, for the horizontal neighbors are usually close by, especially if the 2D mesh is sorted along a space-filling curve.*

*The following text is added:*
*"FESOM2 uses vertical as a tighter index than the horizontal to re-use the information on horizontal neighborhood along the vertical column. This is also favorable for vertical operations and generally does not lead to cache misses in the horizontal, especially if the 2D mesh is sorted along a space-filling curve."*

**• A secondary choice is whether to make a 'super array' for tracers, where you use a tracer index to loop through all the tracer variables, rather than individual tracer variables. If you did this, what order did you place the indices, and why?**

*Concerning the tracers, currently, the implementation is with one array tracer(num_levels, num_2D_nodes, num_tracers). The reason is that advection of tracers is done by applying advection scheme to each tracer separately. This means that outer loop is over tracers. In this case each tracer is contiguous in memory. The other strategy would be to apply advection to all tracers simultaneously, in which case the storage would be tracer(num_tracers, num_levels, num_2D_nodes) or tracer(num_levels, num_tracers, num_2D_nodes), depending on the number of tracers. This second strategy is not pursued. The reason is that it would require allocating more memory for all options that we have in the code because FCT and high-order advection schemes need auxiliary arrays to store*

*intermediate data. Allocating them is not a problem, but cache misses will be much more likely. In our current strategy such, arrays are allocated only for one tracer.*

*We have added the following sentence::*
*"The tracers are stored in a single array tracer(numlevels, 15 num2Dvertices, numtracers), and advection and diffusion computations are applied to each tracer separately in a loop over the tracers. In this case, each tracer is contiguous in memory. Auxiliary arrays needed for FCT and high-order advection are allocated for one tracer only."*

**• Did you consider trying to make your halos wider for less frequent communication? That doesn't work for your implicit SSH solve if it requires a global communication, but it is a possibility in other parts of the code. Would be good to comment on this design choice in the text.**
*The 3D-parts have no communication problem and would not benefit from wider halos. Also the memory-boundedness would likely become worse. The halo exchange is overlapped with computation in many parts of the code.*
*It would be interesting for the ice model and, indeed, for the SSH solver to see if they get faster with larger halos. As for large core counts, the global sums dominate the time used in each iteration, a larger halo and thus a stronger Schwarz preconditioner would reduce the number of iterations at no extra cost. However, a halo of 1 is hard coded in pArms and would be not easy to extend.*

*We have added the following sentence:*
*"On present architectures, a possible strategy to reduce the number of communications in the SSH and sea-ice parts of the code is to make halos wider. "*

**My other comments are very minor. I've attached a copy with grammatical corrections.**
*Thank you very much for grammatical corrections, we have implemented them all.*
**1. Does fArc mean something?**
*Not really, it just FESOM Arctic.*

**2. Fig 7, 9: Add perfect scaling line**
*Done*

**3. Table 3. Label unit for time step**
*Fixed*
* * *
**Interactive Comment (Marshall L. Ward)**

**My main comment is that the numbers which I presented at ECMWF were attempting to assess the scalability of existing configurations of NEMO, MOM5, and MOM6 in use on our system(Raijin at NCI Australia), and were not necessarily representative**

measures of performance of the models. For example, the NEMO configuration was quite old even at the time (v3.4), and two attendees pointed out potential implementation issues of the tripole that were resolved in v3.6. Additionally, I have my own internal numbers showing that MOM5 0.25 can scale much higher than 25 yr/day. For example, I have shown results showing performance of sea ice (SIS) bound timings of 35 yr/day, and can increase this to nearly 60 yr/day, when sea ice is running concurrently to MOM5 and we allow high inefficiency. (This is my usual "reference", a presentation at AMOS in 2016 where the ocean model was individually profiled: https://marshallward.org/talks/amos2016.html#/11/ though again it's just a presentation and hasn't necessarily been stress tested.)

I know how difficult it can be to find cited performance numbers for ocean models, and I have probably not published enough of my own numbers, but feel a bit uncomfortable being associated with these particular numbers and wonder if they could be replaced or revised in some way.
I do have one published paper of MOM performance numbers which may be suitable:
* https://link.springer.com/chapter/10.1007/978-3-319-15994-2_55
And Balaji et al's CPMIP paper may provide some useful comparison numbers:
* https://doi.org/10.5194/gmd-10-19-2017
Or, if you do not want to change the reference, maybe just disclose somewhere that the numbers are not necessarily reflective of peak performance but "typical" performance, in which case I'd request that you use the lower 1000 CPU timings.

*We have updated the numbers by using suggested publication and MOM5 numbers from Kiss et al., 2019.*

**References:**

Smith, B., Bjorstad, P., & Gropp, W. (2004). Domain decomposition: parallel multilevel methods for elliptic partial differential equations. Cambridge university press.